# The Involvement of MicroRNAs in Innate Immunity and Cystic Fibrosis Lung Disease: A Narrative Review

**DOI:** 10.3390/cimb48010058

**Published:** 2026-01-02

**Authors:** Annalucia Carbone, Namra Sajid, Piera Soccio, Pasquale Tondo, Donato Lacedonia, Sante Di Gioia, Massimo Conese

**Affiliations:** 1Department of Clinical and Experimental Medicine, University of Foggia, 71122 Foggia, Italy; annalucia.carbone@unifg.it (A.C.); namra.sajid@unifg.it (N.S.); sante.digioia@unifg.it (S.D.G.); 2Department of Precision Medicine in Medical, Surgical and Critical Areas, University of Palermo, 90127 Palermo, Italy; 3Department of Medical and Surgical Sciences, University of Foggia, 71122 Foggia, Italy; piera.soccio@unifg.it (P.S.); pasquale.tondo@unifg.it (P.T.); donato.lacedonia@unifg.it (D.L.); 4Institute of Respiratory Diseases, Policlinico Riuniti of Foggia, 71122 Foggia, Italy

**Keywords:** cystic fibrosis lung disease, microRNAs, inflammation, immune response, precision medicine

## Abstract

Pulmonary involvement in cystic fibrosis (CF) is characterised by respiratory infections caused by bacteria, viruses, and fungi, as well as by dysregulated inflammatory and immune responses. Although essential for the host’s initial defence against these microorganisms, the innate immune response is altered in its main cellular (airway epithelial cells (AECs), monocytes, macrophages, and neutrophils) and molecular (cytokines, chemokines, signal transduction pathways, and transcription factors) components. MicroRNAs (miRNAs) form a regulatory network at the level of inflammatory and immune responses, and their dysregulation has been observed in immortalised and primary CF AECs as well as in monocytes, macrophages, and neutrophils from CF patients. Although the study of individual miRNAs is helping to dissect the specific altered events in CF lung disease (CFLD), large-scale genomic and transcriptomic studies are more likely to capture its full complexity. The studies we identified suggest that miRNAs are involved in various processes related to CFLD, including impaired pathogen response, compensation for hyperinflammation, altered antigen presentation, and wound healing in AECs and macrophages. However, clinical studies involving large cohorts of patients are needed to obtain meaningful results and identify new therapeutic targets. Equally important will be the study of the miRNome as circulating biomarkers for the purposes of diagnostic and prognostic precision medicine.

## 1. Introduction

Cystic fibrosis (CF) (Online Mendelian Inheritance in Man (OMIM) 219700) is a genetic disease specifically known to present several pulmonary manifestations contributing mostly to high morbidity and mortality among affected individuals [1,2,3]. The hallmarks of CF lung disease (CFLD) are infections by opportunistic pathogens and chronic inflammation [4]. Dysfunction or the lack of the CF transmembrane conductance regulator (CFTR) chloride and bicarbonate channel leads to abnormal hydration and viscoelasticity of airway mucus [5]. This alteration results in mucus obstruction of the airways, creating an environment that promotes infection and chronic inflammation, thereby fuelling a “vicious cycle” [6,7]. Opportunistic bacterial pathogens infesting the CF airways comprise both Gram-positive (*Staphylococcus aureus*) and Gram-negative (*Haemophilus influenzae*, *Burkholderia cepacia*, and *Pseudomonas aeruginosa*) species [8]. Viral infections often superimpose on bacterial infections and are the cause of pulmonary exacerbation during the course of a chronic disease [9,10]. The wound healing process occurring during damage and repair of the CF airway epithelium is intrinsically altered; recurrent infections associated with inflammation in CF airways lead to a cycle of damage and repair of the epithelium without complete reepithelialisation [11,12,13]. Altered repair of the airway epithelium contributes to lung remodelling, resulting in the destruction of proximal airways and ultimately lung function insufficiency [14,15].

The important characteristic of the CFLD inflammatory response is the unsolvable infiltration of neutrophils into the airways, leading to the release of proteases and oxidases that destroy the epithelial surface and stimulate proinflammatory gene expression [6,16,17]. Overexpression of the potent neutrophil chemokine interleukin 8 (IL-8) (also known as CXCL8) in the lungs of CF patients leads to the accumulation of neutrophils that potentiate inflammation [18].

MicroRNAs (miRNAs) are small non-coding RNAs that regulate gene expression at the post-transcriptional level by specifically binding to their cognate mRNAs, inhibiting their translation or inducing their degradation [19]. miRNAs regulate more than 60% of human protein-coding genes, affecting many physiological functions [20]. For this reason, miRNAs play a critical role in many diseases characterised by dysregulation of miRNA expression [21]. In CF, certain studies have focused on the role of miRNAs in regulating CFTR gene expression [22] and the regulation of inflammatory processes [23]. miRNAs are important in regulating innate immune defence and inflammatory processes, as their dysregulation contributes to chronic inflammatory lung diseases such as CFLD [24]. The altered expression of specific miRNAs has been observed in CF [22,23], potentially creating a link between CFTR mutations and the proinflammatory phenotype.

Despite the relevance of neutrophils in damaging the CF airways, other innate immunity cell types, such as monocytes and macrophages, and non-immune cells, such as epithelial cells [25,26], may be involved in the pathogenesis of CFLD. Given the complexity of the cell types involved in CFLD, the role of each immune cell population in miRNA expression and function will be described and interpreted.

## 2. Materials and Methods

We conducted a narrative review with a structured search of PubMed/Medical Literature Analysis and Retrieval System Online (MEDLINE) (https://pubmed.ncbi.nlm.nih.gov (accessed on 1 August 2025)), and Scopus (https://www.scopus.com (accessed on 10 August 2025)) using controlled vocabulary and free-text terms for “cystic fibrosis”, “airway epithelial cells”, “macrophages”, “neutrophils”, “innate immunity”, and “microRNAs”. To gain a deeper insight into the relationship of miRNAs with CF, genomic and transcriptomic studies performed by using immortalised and primary epithelial cells were selected from the Gene Expression Omnibus (GEO) DataSet [27] by using the keywords: “Cystic fibrosis” AND “miRNA Expression” AND “Homo sapiens”. Results were confirmed by interrogating TargetScanHuman [28], miRNet [29], miRTargetLink [30], and miRWalk [31]. No artificial intelligence (AI)-driven analysis, data generation, or study design support was employed.

## 3. Airway Epithelial Cells

Airway epithelial cells (AECs) can be considered an important barrier to mucosal penetration of microorganisms due to their tightness [32]. Moreover, in addition to providing an effective mucociliary clearance of the lower airspace, AECs are relevant players in the immune response to endogenous and exogenous molecular patterns [33,34,35]. Since immune responses have frequently been divided into three types, we will briefly outline each response and the role of AECs in the respective types [33]. The essential feature of type 1 AEC responses is the induction of type I and III interferons (IFNs) and interferon-stimulated genes (ISGs) to control infections by respiratory viruses [36]. Type 1 immunity in AECs is induced in response to pathogen-associated molecular patterns (PAMPs) and damage-associated molecular patterns (DAMPs) through pattern-recognition receptors (PRRs), which activate antimicrobial and proinflammatory responses to effect innate and acquired immunity [33]. Type 2 immunity in AECs is fundamental in controlling multicellular parasite and fungal infection of the airways by upregulating expression of the secreted mucin 5AC (MUC5AC), which traps pathogens in the airway lumen [37]. Recognition of these pathogens occurs through the sensing of chitin, which is found in the cell walls or exoskeletons of organisms such as fungi and parasites [38,39], by the PRR protein LysM domain-containing 3 (LYSMD3), featuring a lysin motif (LysM) as a carbohydrate binding module [40]. AECs can also recognise other fungal cell wall components such as β-glucans, galactomannans, and mannoproteins via C-type lectin-type receptors (CLRs) [41]. Mucus hypersecretion and chitinase production by AECs are upregulated mostly by IL-13, a T helper (Th) 2 cytokine [42,43]. The type 3 AEC response is involved in killing extracellular bacteria and fungi and in inactivating viruses during the extracellular phase of their infectious cycle [33]. Toll-like receptors (TLRs), nucleotide-binding and oligomerisation domain (NOD)-like receptors (NLRs), and complement receptors are involved in sensing PAMPs and DAMPs derived from tissue and cell damage exerted by these pathogens and by the inflammation itself [33]. Cytokines involved in the amplification of these responses are IL-22 and multiple IL-17 family members, which recruit the effector neutrophils [44,45]. Besides neutrophil recruitment, the induction of inflammatory cytokines such as IL-1β, IL-6, and the tumour necrosis factor α (TNF-α), as well as the induction of antimicrobial peptides (AMPs), are the other pillars of type 3 immunity [33].

### 3.1. Dysregulation of Airway Epithelial Cells in Cystic Fibrosis

The dysfunctional state of AECs may derive intrinsically in the absence of infection from *CFTR* mutations, which ultimately leads to an impairment in innate host defence systems [46,47,48,49]. As a primary consequence, CFTR dysfunction results in defective bicarbonate secretion, which in turn reduces the pH of airway surface liquid (ASL), impairing the activity of AMPs and mucus secretion [50].

Innate immune properties of AECs are also disturbed in CF. Regarding the type 1 AECs, the response of primary CF vs. non-CF bronchial epithelial cells (BECs) to rhinovirus (RV) infection was studied [51]. RV serotypes are classified as major or minor groups depending on the surface receptor used to infect target cells [52,53]. Infection (1 hr) with the major group RV16 of primary CF AECs resulted in a trend towards a diminished IFN response in comparison to healthy AECs at the level of IFN-λ1, IFN-λ2/3, and IFN-β, the PRRs TLR3, retinoic acid-inducible gene I (RIG-I), and melanoma differentiation-associated protein 5 (MDA5), as well as ISGs [51]. On the other hand, IFN pathway induction upon the minor group RV1B infection was significantly increased at the level of IFNs and PRRs in CF compared to healthy BECs [51]. Interestingly, the potent IFN inducer polyinosinic:polycytidylic acid (poly(I:C)) stimulated heightened levels of CXCL8/IL-8, IL-6, and CXCL10/interferon gamma-induced protein 10 (IP10), a CXCR3 chemokine critically important in the development of a Th1 response to extracellular pathogens [51]. Previously, Dauletbaev et al. have shown no differences between primary CF and healthy human bronchial epithelial cells (HBEs) in the secretion of the antiviral cytokine IFN-β and the proinflammatory cytokine IL-8, as well as in the expression of the interferon-responsive gene 2′-5′-oligoadenylate synthetase 1 (OAS1) upon infection with RV16 for 24 hr [54]. These data highlight that experimental conditions (short vs. continuous exposure) may change the magnitude of response elicited by CF cells. In line with data obtained previously with RV [51], influenza A virus (IAV) infection of CFTR knockdown (KD) AECs showed robust proinflammatory response, as evidenced by the IL-8, IL-6, and IP-10 gene transcripts [55]. Interestingly, dysregulation of the IFN signalling pathway in CFTR KD AECs was also observed, i.e., downregulation of the interferon regulatory factor 5 (IRF5), which is a member of the IFN regulatory factor (IRF) family, and TLR9, which functions to alert the immune system of viral and bacterial infections by binding to DNA sequences rich in CpG motifs [55]. Overall, these studies highlight defective IFN signalling responses that could contribute to the dysregulated tolerance of AECs against pathogens.

AECs respond to extracellular pathogens through TLR-dependent mechanisms and TLR-independent cytosolic receptors, leading to a type I IFN response [56]. A defective type I IFN signalling in response to *P. aeruginosa* has been reported in CF cells [57]. Using different models of immortalised CF and non-CF pairs of AECs, *P. aeruginosa* was demonstrated to induce less IFN-β, IFN-β-regulated protein CXCL10/IP-10, and type III interferon IFN-λ mRNA and protein expression in CF cells as compared with their non-CF counterparts [57].

The PRRs TLR1–10 and the myeloid differentiation factor 88 (MyD88) are expressed and functional in CF airway epithelia, thereby allowing CF AECs to participate in type 1 and 3 immune modules [58,59,60]. However, some studies deny a higher expression of these immune receptors in CF vs. non-CF cells [59,60]. It was also difficult to detect a global effect of CFTR dysfunction on TLR expression in the lung [59,60]. Rather, it has been reported that the lipopolysaccharide (LPS) PRR protein TLR4 is displayed at very low levels on the apical surface of CF AECs [61,62], possibly altering the response to airway Gram-negative bacteria and contributing to chronic bacterial infection in CF airways. Nevertheless, microbial and viral factors can enhance the expression of TLRs in CF airway epithelia and thereby induce cytokine production [59,63].

Intracellular signal pathways are deregulated in AECs and other immune cells, such as monocytes and macrophages. To this end, CFTR is likely to have an important effect on these pathways both as a chloride channel and as a regulator of various other ion channels, transporters, and receptors [64,65]. Several studies have described intrinsic upregulation of signalling pathways associated with proinflammatory cytokine transcription in CF epithelial cells. The NF-κB pathway is basally hyperactivated in CF primary AECs obtained from nasal polyps, and this dysregulation correlates with increased IL-8 secretion [66]. Additionally, exposure of primary CF AECs to *P. aeruginosa* led to the activation of the NF-κB pathway and downstream expression of IL-8 [67]. Also, the mitogen-activated protein kinase (MAPK)/extracellular signal-regulated kinase (ERK) and activated protein (AP-1) pathways have been implicated in CF AEC inflammatory processes upon incubation with different stimuli [68,69,70,71]. In this context, altered calcium signalling and oxidative stress participate in dysregulation of these pathways in CF AECs [64,65,72]. Furthermore, a defect in the autocrine mechanism leading to the loss of nitric oxide synthase (NOS) 2, a major effector in host defence against viruses and bacteria, was identified, as IFN-γ-inducible factors failed to stimulate NOS2 expression in CF primary AECs [73]. The Akr (mouse strain) thymoma (Akt) pathway may also be defective in CF AECs, as a study demonstrated that pharmacological or genetic inhibition of CFTR in AECs prevents phosphoinositide 3-kinase (PI3K) plasma membrane localisation, which is necessary for Akt phosphorylation [74].

More recently, a systems biology approach using publicly available databases for CF respiratory epithelial cells and their non-CF control counterparts allowed the generation of a CF network that recapitulates immune signalling dysregulation [75]. Pathways that were identified by this strategy were nuclear factor kappa B subunit 1 (NFKB1), NFKB2, v-rel avian reticuloendotheliosis viral oncogene homologue A (RELA), and RELB, all part of the NF-κB complex; FBJ murine osteosarcoma (FOS); and the cellular homologue of the viral oncoprotein v-jun, discovered in avian sarcoma virus 17 and named for *jū-nana* (the Japanese word for 17) (JUN), with FOS being part of the AP-1 complex. Other pathways included caspase (CASP) 3 and 7 as the effectors of apoptosis; CASP1, the effector of pyroptosis, a highly proinflammatory cell death mechanism; regulators of the actin cytoskeleton pathway (ACTN4, ARPC5, PFN, MYL12B, and VCL); and IRF1,3,5, and 7, involved in the innate immune response phenotype controlling the expression of type I interferons upon viral infection [76].

### 3.2. Contribution of miRNAs to Airway Epithelial Cell Dysfunction in Cystic Fibrosis

Several specific miRNAs are involved in the regulation of AEC immune responses in CF (Table 1).

#### 3.2.1. Overexpressed miRNAs

Elevated levels of miR-155 were observed in CF lung epithelial cells and circulating neutrophils, contributing to the inflammation by overexpressing IL-8 [77]. IB3-1 CF immortalised cells presented higher expression levels of miR-155 and let-7c than wild-type CFTR-repaired daughter cells IB3-1/S9, a result that was confirmed ex vivo in primary bronchial AECs and blood neutrophils only for miR-155 [77]. Mechanistically, miR-155 inhibits the translation of Src homology 2 (SH2) domain-containing inositol polyphosphate 5-phosphatase 1 (SHIP1), in turn activating the PI3K/Akt and MAPK signalling pathways, which ultimately leads to stabilisation of IL-8 mRNA and hyperexpression of IL-8 protein [77].

miR-636 is also linked with inflammation in CF. miR-636 was found to be significantly overexpressed in CF air–liquid interface (ALI) cell cultures compared to non-CF cultures [78]. In a subsequent study in CF primary bronchial cells differentiated in ALI cultures, it was found that miR-636 overexpression by a mimic miRNA repressed the interleukin-1 receptor 1 (IL1-R1) and inhibitor of nuclear factor kappa B kinase subunit beta (IKK-β), and increased TNF receptor sub-family 11A (TNFRSF11A) (also called receptor activator of nuclear factor kappa B (RANK)), protein expression in CF cells, resulting in overall decreased activation of the NF-κB pathway [79]. Moreover, the miR-636 mimic decreased the secretion of IL-8 and IL-6 compared to the mimic control in CF cells. In addition, a significant increase in miR-636 expression in CF neutrophils compared to non-CF neutrophils was found.

A deep sequencing study of miRNome found that miR-181a-5p was upregulated in CF ALI cultures obtained from bronchial brushings, nasal brushings, and nasal polyps [80]. Insulin-like growth factor 1 (*IGF1*) and WNT1-inducible signalling pathway protein 1 (*WISP1*) were predicted to be targeted by miR-181a-5p and were found to be downregulated in CF cells. A miR-181a-5p inhibitor led to an increase in the mRNA levels of *IGF1* and *WISP1*, a key component of cell proliferation and migration programmes, confirming that miR-181a-5p may play a role in their deregulation. On the other hand, miR-101-3p and its isoforms were also deregulated in CF cells. miR-101-3p and miR-181a-5p silencing stimulated increased WISP1 mRNA and protein levels, as well as wound healing [80].

#### 3.2.2. Downregulated miRNAs

miR-126 was found to be significantly decreased in CF bronchial brushings compared with controls [81]. miR-126 targeted the target of Myb1 (TOM1) and the Toll-interacting protein (Tollip, which form a complex that has been proposed as a negative regulator of IL-1β- and TNF-α-induced signalling pathways [84]. CF bronchial brushings and CF cell lines showed upregulation of TOM1 and Tollip as compared to their non-CF counterparts [81]. Moreover, overexpression of miR-126 let to decreased TOM1 protein levels, while knockdown of TOM1 increased IL-8 protein production in response to LPS, IL-1β, or lipopeptide in CFBE41o– cells. Linked altogether, these data suggest a control role of miR-126 in lung inflammation, accompanied by the increased expression of TOM1 that may play an anti-inflammatory role in the CF lung to compensate for the high proinflammatory burden.

miR-17 was significantly decreased in vivo in CF bronchial brushings, while overexpression of miR-17 in CF AECs of tracheal and bronchial origin reduced the level of IL-8 protein under basal conditions and upon stimulation with LPS, *Pseudomonas*-conditioned medium, or bronchoalveolar lavage fluid (BALF) obtained from CF patients [82]. In the two studies previously mentioned [81,82], bronchial brushings were not processed to obtain pure suspensions of AECs, and in principle, a contamination with immune cells would occur. However, similar results obtained with immortalised CF AECs strongly suggested that miRNA expression was related to AECs.

Downregulation of miR-199a-3p was observed in primary ALI-cultured BECs from CF patients and non-CF controls [78]. The study of CF bronchial explants confirmed miR-199a-3p downregulation as compared to the controls and that the low expression of miR-199a-3p was inversely proportional to high expression of IL-8 and IKKβ, thus contributing to chronic pulmonary inflammation [78]. In the CF cell line CFBE41o−, a miRNA mimic suppressed the NF-κB pathway by directly targeting *KBKB*, which encodes IKKβ, and lowered IL-8 secretion [78]. CF lung tissues are a mixed composition of non-immune cells (i.e., AECs) and immune cells, as well as other cell types (fibroblasts and endothelial cells) [85]. Thus, regulation of miR-199a-3p in lung explants would not warrant specific AEC expression. Nevertheless, experiments in primary BECs confirmed that miR-199a-3p is downregulated in the CF airway epithelium [78].

miR-93, which is normally expressed at high levels under basal conditions, was downregulated during *P. aeruginosa* infection of CF immortalised cell lines IB3-1 and CuFi-1, correlating with stabilisation of IL-8 mRNA and increased levels of IL-8 expression [83].

## 4. Resident and Recruited Lung Macrophages

The two major macrophage populations of the lung are those located in the airway lumen, i.e., alveolar (AMs), and those residing in the lung parenchyma, i.e., interstitial (IMs) [86]. AMs are considered the sentinels of lung health. They remove cellular and extracellular matrix debris and recycle surfactant molecules, maintaining immunological and physiological homeostasis [86]. Moreover, they respond in a suppressive manner to antigens to prevent damage to alveoli and disruption of the lung structure without triggering neutrophil influx and subsequent excessive inflammation [86,87,88]. AMs exert their protective effects via non-specific lines of defence such as high phagocytic ability and the secretion of antimicrobials and NO, as well as chemokines and cytokines TNF-α, IL-1β, and IFN-γ that recruit and activate neutrophils, monocytes, and dendritic cells (DCs) [89,90]. As AMs initiate inflammation, they are also responsible for resolving inflammation through the secretion of immunoregulatory cytokines such as TGF-β, IL-1RA, and prostaglandins [91,92,93]. IMs perform homeostatic and immunoregulatory functions [86]. It has been suggested that they are required for the maintenance of lung nonhematopoietic cells, although the precise mechanism is not known [94]. IMs are thought to have immunoregulatory function within the lung tissue by their propensity to produce cytokines such as IL-10, TGF-β, and IGF1, as well as reactive oxygen scavengers such as SOD2 [86,95,96,97].

Pulmonary macrophages can exist under different phenotypes, whose extremes are called M1 (or “classically activated macrophages”) and M2 (or “alternatively activated macrophages”) [88,98]. M1 macrophages participate in type 1 immune response such that they are activated by IFN-γ, LPS, and TNFα, produce proinflammatory cytokines, are involved in direct destruction of intracellular pathogens, and promote a local Th1 environment [88]. M2 macrophages are characterised by their participation in type 2 immune responses and thus arise in response to IL-4 and IL-10 [99]. A subset of M2 macrophages exerts different functions. M2a facilitates parasite encapsulation and destruction, M2b is immunoregulatory, and M2c is involved in tissue remodelling and matrix deposition [100]. Owing to their extremely plastic architecture, pulmonary macrophages tailor their phenotype to meet the immediate immunological needs [88].

The PI3K/Akt pathway is rapidly activated upon LPS stimulation and exerts multifaceted regulatory effects on innate immune functions, influencing macrophage differentiation, polarisation, and survival [101]. One of its key roles is the negative regulation of TLR4 signalling, which serves to prevent excessive inflammation and promote immune homeostasis [102,103]. Activation of PI3K leads to phosphorylation of Akt, which in turn attenuates downstream proinflammatory cascades, including the NF-κB and MAPK pathways, thereby limiting the production of cytokines such as TNF-α, IL-6, and IL-1β [103]. This inhibitory function is closely linked to the capacity of PI3K/Akt to modulate the formation and activity of the TLR4/MyD88 signalling complex, highlighting an intricate crosstalk between these pathways [102]. Collectively, these studies position the PI3K/Akt pathway as a crucial negative feedback mechanism in TLR4-mediated signalling, fine-tuning the balance between immune activation and resolution of inflammation.

Besides tissue-resident macrophages, neutrophils and monocytes are attracted into the lung under inflammatory conditions [104]. In general, phagocytic cells cooperate during the onset, progression, and resolution of inflammation [105]. Upon recognition of DAMPs and PAMPs, resident macrophages and DCs are activated and produce proinflammatory cytokines and chemokines, including TNF-α, IL-6, CXCL8, and 12-hydroxyeicosatetraenoic acid (12-HETE), which stimulate the recruitment of neutrophils to the injured lung [106]. After neutrophils, circulating classical Ly6C^+^ monocytes migrate into the lung in a C-C motif chemokine receptor 2 (CCR2)-dependent manner in response to chemokines such as the C-C motif chemokine ligand 2 (CCL2) (also known as MCP-1) and CCL7 (MCP-3) [107]. These recruited macrophages can either acquire an AM-like phenotype or are phenotypically distinct, providing functional roles in epithelial repair, induction of fibrosis, or enhanced inflammation [108,109,110,111,112]. During lung infection or other inflammatory insults, the diversity of macrophage populations expands and populations of recruited macrophages replace or join the pool of resident AMs and IMs [86]. Once neutrophils and monocytes have entered into the lung, a set of ‘brakes’ prevent further infiltration of leukocytes [113], inflammation-resolving mediators are produced [114], efferocytosis of apoptotic neutrophils by lung macrophages occurs [115], and homeostasis is regained.

### 4.1. Dysfunction of Macrophages in Cystic Fibrosis

Several studies have shown that CF macrophages contribute directly to lung hyperinflammation and disease progression [116,117,118]. Above all, CF macrophages present altered phagocytosis [119] and decreased killing of lung pathogens such as *P. aeruginosa* and *B. cepacia* [120,121,122]. In the context of complement-mediated opsonisation [123], reduced expression of integrin αM (cluster of differentiation 11b (CD11b)) and dysfunctional autophagy [124] have been reported to contribute to defective bacterial killing by CF macrophages. Treatment of human AMs isolated from CF patients with IGF1, a potent Akt activator, enhances their capability to kill *P. aeruginosa* [122], implying a deregulation of the PI3K/Akt pathway in CF macrophages.

Aberrant macrophage behaviour in CF has been associated with exaggerated inflammatory responses to bacterial stimuli [125]. In CF macrophages derived from peripheral blood mononuclear cells (PBMCs), secretion of the soluble endotoxin receptor CD14 (sCD14), IL-1β, IL-6, and TNF-α is heightened, while expression of CD11b and TLR-5 is profoundly decreased, highlighting that CF macrophages are unable to recognise pathogens and contribute to an unremittingly inflammatory state [126]. Some studies showed that hypersecretion of proinflammatory cytokines in CF macrophages is due to reduced levels of the scaffold protein caveolin 1 (CAV1) that induces TLR4 signalling [127,128]. CAV1 binds to and negatively regulates TLR4 in macrophages [129] and this protective mechanism is defective in CF macrophages [128].

Upon lung damage, classic Ly6C^hi^CCR2^+^ monocytes (cMons) are recruited from the circulation in a CCR2-dependent manner [107,130]. High numbers of CCR2^+^ monocytes have been found in submucosal areas of lung explants from patients with CF [131]. CCR2^+^ monocyte recruitment and monocyte-derived macrophage counts were increased in CF mouse lungs in response to chronic LPS exposure, which would sustain a proinflammatory environment by facilitating neutrophil mobilisation [131]. Finally, CCR2^+^ cMons and cMon-derived macrophage populations contribute to pathogenic TGF-β levels and Sma- and Mad-related protein 2 (SMAD2)-dependent signalling in CF murine lungs exposed to chronic inflammatory triggers, associated with severe lung damage with enlarged alveolar spaces and distorted alveolar structure [131].

Macrophages are involved not only in the initiation (M1) but also in the resolution of lung inflammation (M2) [132]. A defective M2 polarisation in CF PBMC-derived macrophages was observed, whilst M1 polarisation was not affected, pointing to defective macrophage function in the resolution phase of inflammation in CF [133]. This polarisation imbalance, together with the suppression of M2 macrophage polarisation in patients with CF, was confirmed by other studies [134,135]. It has been suggested that the reduced polarisation of CF M2 macrophages could stem from increased levels of ER stress [134], but no convincing evidence has been provided yet.

Thus, CF macrophages exhibit several types of cell-autonomous immune dysfunction that contribute to CFLD and render macrophages potential targets for effective CF treatments.

### 4.2. Contribution of miRNAs to Macrophage Dysfunction in Cystic Fibrosis

miR-199a-5p is upregulated in CF macrophages derived from murine bone marrow and human peripheral blood and in the lungs of CF mice exposed to inflammatory triggers [136]. By targeting *CAV1*, the elevated expression of miR-199a-5p promotes CF-associated lung hyperinflammation through disruption of CAV1-mediated inhibition of TLR4 signalling [136]. This axis is regulated by PI3K/Akt signalling, which is dysfunctional in CF macrophages, leading to increased levels of *miR-199a-5p* and decreased levels of *CAV1* [136]. In summary, CF macrophages display blunted PI3K/pAkt signalling in response to TLR4 activation, which leads to the accumulation of miR-199a-5p and decreased CAV1 expression, ultimately leading to ineffective negative feedback of TLR4 signalling and hyperinflammation [136]. Following miRNA profiling in PBMC-derived macrophages from CF and non-CF individuals, a panel of differentially expressed miRNAs in CF macrophages, as compared to non-CF macrophages, was identified [137]. Among these, miR-146a was associated with significant enrichment of validated target genes that are involved in responses to microorganisms and inflammation. miR-146a was found to be upregulated in CF macrophages, and accordingly, the TNF receptor-associated factor 6 (TRAF6), a crucial adaptor for TLR-mediated NF-κB signalling, was downregulated. Moreover, inhibition of miR-146a reduced IL-6 production in LPS-stimulated CF macrophages [137]. Thus, miR-146a is functional and contributes to limiting IL-6 production in CF macrophages.

A microarray profiling of X-linked miRNAs was performed in peripheral blood CD14^+^ monocytes from the CF and non-CF study populations [138]. Of the 86 investigated miRNAs, miR-224-5p (3.0-fold) (*p* = 0.1336) and miR-452-5p (1.9-fold) (*p* = 0.1242) showed the greatest increase in the CF cohort as compared to non-CF subjects. SMAD4, a validated target of miR-244-5p, was downregulated in CF monocytes. SMAD4 and miR-224-5p levels were negatively correlated in CF monocytes, although not significantly [138]. SMAD4, a transcription factor activated by TGF-β, is required for endotoxin tolerance in monocytes, a mechanism to control hyperinflammation [139].

## 5. Dysregulation of Neutrophils in Cystic Fibrosis

As recalled above, neutrophils are the first line of innate immune cells occurring during sterile and non-sterile lung inflammatory conditions and recruited from the blood [140,141]. Despite their recruitment for fighting infections, CF neutrophils show various abnormalities, such as protease secretion, the production of reactive oxygen species, migratory responses, cell-to-cell clustering, microbe containment, and the formation of neutrophil extracellular traps (NETs) [16,142,143,144].

Some studies have reported that neutrophils die prematurely in CF airways via either secondary necrosis following apoptosis and the lack of macrophage efferocytosis, or via cell death accompanying the formation of NETs [145,146]. Others have shown that CF neutrophils have a primary defect causing decreased spontaneous apoptosis, allowing increased levels of NET formation that can promote inflammation [147]. In line with this observation, other investigations have pointed out that airway neutrophils remain viable upon entry into the CF lung, displaying a metabolic adaptation featuring primary granule release, immunomodulatory activity, and metabolic licensing termed “GRIM” [148,149,150]. GRIM neutrophils undergo active exocytosis of primary granules, leading to a massive release of elastase and myeloperoxidase, which damage the airway tissue and perpetuate inflammation [149], upregulate the cAMP-responsive element-binding protein (CREB) and mTOR prosurvival pathways, and show augmented cell surface nutrient transporter expression and glucose uptake [148].

### Contribution of miRNAs to Circulating Innate Immune Cell Dysfunction in Cystic Fibrosis

miR-636 was found be upregulated in CF neutrophils but not in the plasma [79]. Whether miR-636 overexpression can deregulate neutrophil functions, such as by decreasing protein expression of IL1-R1 and IKKβ, as well as increasing the expression of RANK, as shown in CF AECs [79], has to be demonstrated. Nevertheless, miR-636 might be considered a biomarker of inflammation in CF blood neutrophils.

A study focused on whether blood leukocytes from male and female CF patients exhibit a differential expression profile of X-linked miRNAs with an impact on the inflammatory process [151]. A significant increase in the expression of miR-223-3p, miR-106a-5p, and miR-221-3p in both CF males and females compared to their sex-matched controls was found. miR-221-3p was significantly higher in CF females than in CF males, with a corresponding lower expression of the *PDZ and LIM domain 2* (*PDLIM2)* and *suppressor of cytokine signalling 1* (*SOCS1*) mRNA targets. Moreover, a significant positive correlation was observed between miR-221-3p and the *IL-1β* transcript, a finding consistent with the proinflammatory role of this miRNA [151].

## 6. Genomic and Transcriptomic Studies

In order to obtain a deeper insight into the relationship of miRNAs with CF, various genomic and transcriptomic studies have been performed by using immortalised and primary epithelial cells selected from the GEO DataSet [27] by using the keywords “Cystic fibrosis” AND “miRNA Expression” AND “Homo sapiens”. Of the 67 studies found, 4 that dealt with AECs were extracted (Table 2).

The target of miR126-3p, the interleukin 6 cytokine family signal transducer (IL6ST), represents the gp130 subunit of the receptor for IL-6 [154]. Interestingly, it has been recently shown that a CF bronchial cell line is hyper-responsive to both the classic IL-6 and trans-signalling via the soluble form of IL-6Rα (sIL-6Rα) as compared to non-CF counterparts [155]. IL-6-mediated trans-signalling also operates at the neutrophil level, likely perpetuating inflammation [155]. However, IL-6 is also anti-inflammatory [156]. Oglesby et al. [81] demonstrated that CF cells show increased mRNA expression levels of *TOM1*, a negative regulator of the NF-κB pathway, as a consequence of miR-126 downregulation. This effect on TOM1 expression possibly exerts a compensation for the hyperinflammation phenotype occurring in CFLD. Similarly, the upregulation of *IL6ST* by miR-126 downregulation is in line with the attenuation of inflammation. In this work, freshly isolated bronchial brushings were used, likely containing a mixed cell type composition comprising immune cells, indicating that miR-126 downregulation could occur either in immune cells or that immune cells could exert a regulatory effect on AECs [81].

miR-223-3p and miR-145-5p are upregulated in CF bronchial AECs depleted of blood cells and not only suppress the mRNA and protein levels of CFTR but also regulate inflammation and immune responses [153]. The target of miR-223-3p, *IL6ST*, should then be downregulated, attesting to a complex regulation of inflammatory signals between AECs and neutrophils [157]. Interestingly, CF BALF-derived exosomes present elevated levels of miR-145-5p [158] and induce a significant increase in miR-145-5p expression in CFBE41o- and Cufi-1 cells [153]. This effect exemplifies a complex interaction of extracellular vesicles (EVs) present in respiratory secretions with the plethora of cell types of the airways.

Catellani et al. [152] showed that five miRNAs were upregulated (miR-155-5p, miR-370-3p, miR-886-5p, miR-10b-3p, and miR-577-5p) and one (miR-1257) was downregulated in the CFBE41o- and IB3-1 cell lines. Of these, miR-155-5p is involved in inflammation in CF [159]. BTB domain and CNC homologue 1 (BACH1), a target of miR-155-5p, is a transcription factor and a master regulator of antioxidant systems [160]. It controls the expression of CFTR in airway epithelia by either the direct occupancy of *cis*-regulatory elements (CREs) or modulating expression under oxidative stress [160]. BACH1 downregulation is involved in the lower expression of CFTR [161]. On the other hand, BACH1 downregulation by the upregulation of miR-155-5p in airway epithelia would cause an impaired antioxidant system with the increased expression of nuclear factor erythroid 2-related factor 2 (Nrf2) and the cytoprotective enzyme heme oxygenase 1 (HO-1), leading to reduced hypoxia injury [162].

Bhattacharyya et al. [77] found that miR-155-5p is overexpressed in F508del CF lung epithelial cell lines and in samples from CF patients (bronchial brushings and blood neutrophils). miR-155-5p overexpression impaired the activity of SHIP1, which dysregulated the PI3K/Akt signalling pathway by enhancing IL-8 expression. Activation of the PI3K/Akt signalling system attracts immune cells likely to enhance inflammation with upregulated expression of the IL-8 chemokine [77].

Pommier et al. [80] showed that in three CF models, ALI cultures of bronchial, nasal, and nasal polyp cells, impairment of CFTR expression is associated with upregulation of miR-181a-5p and miR-101-3p. Functional analyses indicated that both miR-181a-5p and miR-101-3p repress the expression of WISP1, a secreted matricellular protein implicated in epithelial repair and remodelling, thus linking CFTR deficiency to altered wound healing programmes [80]. These miRNAs are predicted to target the key regulators of inflammation, such as IGF1 [80]. Direct evidence for *IGF1* as a downstream target in this axis within the CF lung tissue remains limited. However, the broader literature demonstrates that patients with CF exhibit both systemic and local deficiency of IGF1 [163], which is associated with impaired AM function and particularly reduced bactericidal activity against *P. aeruginosa* [122]. This finding suggests that compromised innate immunity in CF lungs is not solely due to epithelial defects but also involves immune cell dysfunction driven by low IGF1 protein expression [122,164].

Based on the miRNAs reported in the last section (miR-126-3p, miR-223-3p, miR145-5p, miR-155-5p, miR-181a-5p, and miR-101-3p), we determined interaction with target genes as predicted by the miRTargetlink database [30] and built a miRNA–target gene network (Figure 1). In this network, we selected the most experimentally validated target genes and the most predicted ones. As the network showed, there were diverse targets for each miRNA, and one gene was also targeted by more than one miRNA. miR-126-3p targets *TOM1*, while *IL6ST* is targeted by miR-223-3p, which was verified by the miRNet and TargetHumanScan databases. *BACH1* and the inositol polyphosphate-5-phosphatase D (*INPP5D*)/*SHIP1* genes are targeted by miR-155-5p, which was also validated using the TargetHumanScan and miRNet databases. In the network tree, miR-181a-5p and miR101-3p do not display any interaction with any of the target genes of interest in this database network. miR-145-5p, miR-223-3p, and miR-101-3p directly target the *CFTR* gene and contribute to its expression.

A schematic outline of interactions between miRNAs of interest and genes involved in inflammation, immune response, and wound healing was finally drawn (Figure 2). In this scheme, miR-145-5p is no longer present, since it did not have any interaction with target genes, based on miRTargetLink (Figure 1). On the other hand, we thought it was important to show other connections that were missed in the interaction network of Figure 1; hence, we included the interaction of miR-181a-5p and miR-101-3p with WISP1, as presented by Pommier et al. [80], and the connection between miR-126-3p and IL6ST [154]. Moreover, we also show that miR-155-5p, besides the interactions depicted in the network of Figure 1, affects CFTR expression [77].

## 7. MicroRNAs in Biological Fluids in Cystic Fibrosis Patients

Few studies have focused on miRNAs in plasma derived from CF subjects (Table 3).

Using microarray profiling, Ideozu et al. [165] identified eleven extracellular circulating (EC) miRNAs that differed significantly between CF and healthy control (HC) plasma samples. Among the 2,505 targets, the top target list was dominated by genes involved in miRNA biogenesis and gene regulation. The top significant canonical pathways were primarily associated with signal transduction, including the PI3K/Akt, Wnt/β-catenin, glucocorticoid receptor, and mTor signalling pathways. Although interesting for understanding dysregulated pathways in CF, the results of this study ought to be further corroborated by further prospective investigations and using larger sample sizes, thereby encouraging investigation of the utility of EC miRNAs as biomarkers for CF and its phenotypes.

Personalised medicine is the next big frontier in human disease and is also pursued in CF by tailoring medication regimens to individual genetic profiles [168]. Along this line, a gender gap exists in CF, where females are at a clinical disadvantage compared to males [169]. Interestingly, microRNA expression profiles identified over 100 miRNAs that can be reliably quantified in paediatric CF plasma samples, with the notable result that miR-885-5p is elevated in plasma from girls versus boys under the age of six with CF [166]. The largest clusters of miR-885-5p target genes were formed from the gene ontology biological processes involving tyrosine kinase signalling, cell migration, and motility. miR-885-5p may therefore be potentially useful in monitoring the course of the disease and the CF gender gap, an application towards precision medicine in CF.

EVs are thought to be involved in CFLD pathogenesis [170]. Their enrichment in miRNAs positions them as the extracellular carrier of miRNAs in extracellular matrices. For example, as a start to understanding whether extracellular miRNAs can potentially identify a pulmonary exacerbation (PE) in CF, EV-derived miRNA in sputum, exhaled breath condensate (EBC), and serum were studied in paediatric patients during PE and the stable stage of CF [167]. Since four peripheral and sputum miRNAs (let-7c, miR-16, miR-25-3p, and miR-146a) were the best predictive model of exacerbation, EV-miRNAs contained in peripheral blood or local samples (sputum but not EBC) might have a diagnostic potential in predicting exacerbation in paediatric CF [167].

## 8. Compartmental Distribution of Cystic Fibrosis-Associated miRNAs

A comparative overview of CF-associated miRNAs across cellular and extracellular compartments is provided in Table 4. The analysis highlights specific overlaps among miRNAs previously described as CF-specific, CF-typical, or functionally linked to CF pathophysiology. Both miR-155-5p [77] and miR-636 [79] were detected in AECs and neutrophils. For miR-155-5p, the chronic inflammatory characteristic of the CF airway microenvironment activates NF-κB signalling, driving its upregulation in epithelial cells and neutrophils. Increased miR-155 suppresses SHIP1 expression, thereby enhancing PI3K/Akt activation and contributing to the excessive production of proinflammatory mediators such as IL-8 [77]. miR-636 similarly shows elevated expression in BECs and in circulating neutrophils from individuals with CF, likely reflecting a shared response to persistent inflammatory stress [79]. This pattern suggests a potential role for miR-636 in coordinating inflammatory signalling across epithelial and innate immune compartments. miR-146a was shared between monocyte-derived macrophages [137] and the EC miRNA fraction [165]. miR-146a is a key regulator of innate immune signalling, particularly in monocytes and macrophages, where it negatively regulates NF-κB-dependent proinflammatory responses by targeting TRAF6 and interleukin-1 receptor-associated kinase 1 (IRAK1) [171]. Under chronic inflammatory conditions, miR-146a can be actively secreted into the extracellular space via exosomes or microvesicles, contributing to its detection in circulating EC-miRNAs and mediating intercellular modulation of inflammation [172]. miR-223-3p was found in both AECs [153] and neutrophils [157]. miR-223-3p is highly expressed in mature neutrophils, reflecting its role in myeloid differentiation and neutrophil function [173]. Activated neutrophils can release miR-223 via microvesicles or exosomes, which can then be taken up by AECs [174]. Consequently, miR-223-3p becomes detectable in epithelial cells, indicating a neutrophil–epithelial intercellular miRNA transfer that may help regulate airway inflammation and epithelial responses [174].

## 9. Conclusions and Future Outlook

The complexity of CFLD is compounded by the complexity of the miRNome and the interactions between miRNAs and their molecular targets, making it extremely difficult to develop a model of inflammation and immunity in CF based on their expression. From the appraisal of all the works we scrutinised in this review, it appears that a cell network based on miRNA- or EV-based communication would be possible; however, the cell types considered are single-origin, either AECs, monocytes, macrophages, or neutrophils. Nevertheless, it is tempting to speculate that these cell types involved in CFLD could exchange miRNAs via microvesicles or exosomes, a topic to be best explored in vivo on freshly obtained samples from the airways or in vitro in co-culture systems. More large-scale genomic and transcriptomic studies are needed in order to obtain meaningful results that provide an overall picture of the roles of AECs, monocytes, macrophages, and neutrophils. Furthermore, studies on miRNAs conducted in blood and sputum samples need to be better contextualised. By revealing the molecular complexity of CF, comprehensive genomic profiling can help accelerate translational research and inform future diagnostic and treatment strategies in CFLD.

The exploration of the miRNome in CF could be of extreme importance for diagnostic and prognostic purposes, as well as to gauge the efficacy of therapies. The introduction of CFTR modulators in clinics has been a milestone in CF therapy, effectively restoring CFTR function in a subgroup of patients, but little is known regarding improvement in the innate immune response [175]. miRNA profiling in cells, tissues, and blood-borne samples may provide a deeper insight into gauging these etiological treatments.

The study of the miRNome in innate immune cells (monocytes, macrophages, and neutrophils) and non-immune cells (airway epithelial cells) derived from CF individuals, as well as that of circulating and respiratory miRNAs, is well-positioned not only to advance our knowledge about CF pathophysiologic mechanisms but also for the purposes of diagnostic and prognostic precision medicine in CF.

## Figures and Tables

**Figure 1 cimb-48-00058-f001:**
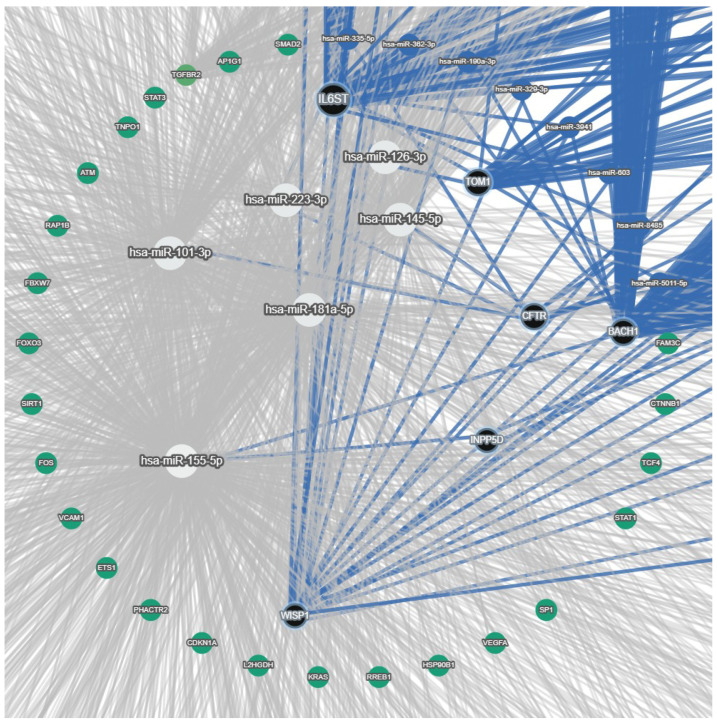
miRNA—target gene network highlighting miRNAs, their validated target, and additional network context. White circle nodes: miRNAs of interest (miR-126-3p, miR-223-3p, miR-145-5p, miR-155-5p, miR-181a-5p, and miR-101-3p). Black circle nodes: target genes of miRNAs of interest (*TOM1*, *IL6ST*, *BACH1*, *INPP5D*/*SHIP1*, *WISP1*, and *CFTR*). Blue circle nodes: other miRNAs in the network. Green circle nodes: other diverse target genes of miRNAs. Blue edges: interaction among precise miRNAs and their specific targets (validated or top predicted targets). Grey edges: background and predicted interactions in the full network.

**Figure 2 cimb-48-00058-f002:**
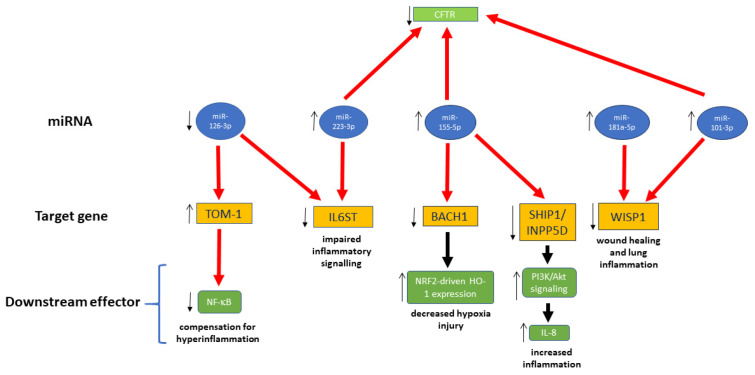
miRNA interactions with target genes involved in inflammation, immunity response, and wound healing. Thick red arrow, downregulation; thick black arrow, upregulation; thin arrows, regulation in CF cells.

**Table 1 cimb-48-00058-t001:** Effects of miRNAs on the inflammatory response in CF airway epithelia.

miRNA Levels Found in CF	Model	Molecular Target	Signal Pathways	Cytokines/Chemokines	Effect on Inflammation (I)/Wound Healing (WH)	References
↑miR-155	IB3-1 cells	↓SHIP1	↑PI3K/Akt and MAPK	↑IL-8	↑I	[77]
↑miR-636	CF primary bronchial cells	↓IL-1R1, IKK-β, RANK	↓NF-κB	↓IL-8, IL-6	↓I	[78,79]
↑miR-181a-5p	CF ALI cultures of AECs from bronchial brushings, nasal brushings, and nasal polyps	↓IGF1 and WISP1	PI3K–Akt deregulation	ND	↓WH	[80]
↓miR-126	CF bronchial brushings, CF immortalised AECs	↑TOM1 and Tollip	↓NF-κB	↓IL-8	↓I	[81]
↓miR-17	CF bronchial brushings, CF immortalised AECs	ND	ND	↓IL-8	↓I	[82]
↓miR-199a-3p	CF bronchial explants, CF primary bronchial AECs, CFBE41o− cell line	↑IKK-β	↑NF-κB	↑IL-8	↑I	[78]
miR-93 ↓upon infection	CF immortalised cell lines IB3-1 and CuFi-1	ND	ND	↑IL-8	↑I	[83]

Downward-pointing arrow, downregulated expression; upward-pointing arrow: uperegulated expression. AECs, airway epithelial cells; Akt, Akr (mouse strain) thymoma; ALI, air–liquid interface; CF, cystic fibrosis; IGF1, insulin-like growth factor 1; IKK-β, inhibitor of nuclear factor kappa B kinase subunit beta; IL-1R1, interleukin-1 receptor 1; IL-6, interleukin 6; IL-8, interleukin-8; MAPK, mitogen-activated protein kinase; ND, not determined; NF-κB, nuclear factor kappa B; PI3K, phosphoinositide 3-kinase; RANK, receptor activator of nuclear factor kappa B; SHIP1, Src homology 2 (SH2) domain-containing inositol polyphosphate 5-phosphatase 1; Tollip, Toll-interacting protein; TOM1, target of Myb1; WISP1, WNT1-inducible signalling pathway protein 1.

**Table 2 cimb-48-00058-t002:** Genomic and transcriptomic studies of miRNAs in CF AEC ex vivo and in vitro models.

Study	Method	Cell Line/Cell Culture	Results	Interpretation
Oglesby et al., 2010 [81]	qRT-PCR and miRNA expression profiling	CF and non-CF bronchial brushings utilised for miRNA profiling	Of the 667 different miRNAs investigated, 93 were found differently expressed in CF patients as compared to controls, 56 were downregulated, while 36 were upregulated, in at least 3 CF patients. miRNA-126 was downregulated in 4 out of 5 CF patients, acting as a proinflammatory agent to increase its target gene TOM1 expression	This study showed that miR-126 is the key regulated miRNA in CF, whose low expression increases the TOM1 level, contributing to innate immune response and proinflammatory characteristics of CF
Catellani et al., 2022 [152]	Microarray expression profiling	Immortalised CF AECs CFBE41o- and 1B3-1, immortalised non-CF AECs 16HBE14o-	Among the 511 target genes of 41 dysregulated miRNAs, the authors evidenced IL-11, IL6R, and IL-18 genes, which are related to lung and airway epithelia. Five miRNAs were upregulated (miR-155-5p, miR-370-3p, miR-886-5p, miR-10b-3p, and miR-577-5p) and one miRNA (miR-1257) was downregulated in both CFBE41o- and IB3-1 cells	Malfunction of CFTR ascribed to the disruption in miRNA expression that contributes to dysregulated biological processes and airway inflammation
Pommier et al., 2021 [80]	Small RNA sequencing	ALI-polyps (CF vs. NCF); ALI-nasal (CF vs. NCF); ALI-bronchial cells (CF vs. NCF); immortalised CF AECs CFBE41o-, immortalised non-CF AECs BEAS-2B and 16HBE14o-	Three ex vivo models were analysed compared to the control for deregulated miRNA expression, with the results demonstrating upregulated miR-181a-5p and members of the miR-449 family. Sequence variants of miR-101-3p, along with miR-181a-5p, directly impact the regulation of WISP1, which causes cell proliferation, migration, and wound healing	Airway epithelia of CF patients have altered miRNAs, including isomiRNAs, that are likely related to disease-relevant phenotypes by modulating their target *WISP1*. The study identifies the mechanistic link between CFTR dysfunction and aberrant wound healing, proposing a new therapeutic target
De Santi et al., 2020 [153]	miRNA sequencing and miRNA profiling	CFBE41o- cell line transfected with wild-type or F508del-CFTR Primary BECs obtained from CF and non-CF subjects, purified and used for miRNA profiling	Several miRNAs (e.g., miR-145-5p, miR-223, miR-494, and miR-509-3p) were overexpressed in AECs of CF patients that bind to the CFTR- UTR3′ region, resulting in suppression of CFTR mRNA and protein levels. miR-223-3p and miR-509-3p were significantly increased in the ALI culture of CFBE41o- cells stably transfected with F508del versus wild-type CFTR	Specific target site blockers of the *CFTR 3′* untranslated region were used to reverse miRNA-mediated inhibition of CFTR. This study suggests a new therapeutic approach of using mRNA–miRNA interaction to increase the CFTR expression

AECs, airway epithelial cells; ALI, air–liquid interface; BECs, bronchial epithelial cells; CF, cystic fibrosis; CFTR, CF transmembrane conductance regulator; IL6R, interleukin 6 receptor; IL-11, interleukin-11; IL-18, interleukin-18; isomiRNAs, isoforms of a reference miRNA; NCF, non-CF; qRT-PCR, quantitative real-time PCR; TOM1, target of Myb 1; WISP1, WNT1 inducible signalling pathway 1.

**Table 3 cimb-48-00058-t003:** miRNAs in biological fluids in cystic fibrosis patients.

Study	Biological Fluid	CF patients	Method	Outcome
Ideozu et al. [165]	Plasma	Discovery phase, 5 CF patients (16.6 ± 4.8 years) and 5 HC (23 ± 1.6 years). Validation phase, 5 patients (22.3 ± 4.4 years) and 5 healthy controls (22.6 ± 3.5 years).	miRNA microarray profiling	Eleven miRNAs differentially expressed between CF and HC samples. The overexpressed miRNAs included hsa-miR-486-5p, 3 family members of let-7 (hsa-let-7b-5p, hsa-let-7c-5p, and hsa-let-7d-5p), hsa-miR-103a-3p, and other miRNAs, while hsa-miR-598-3p was underexpressed in CF
Mooney et al. [166]	Plasma	Six males and six females (age range: 1–6 years, median age: 3.8 years)	miRNA microarray profiling	Two significantly differentially regulated miRNAs in male versus female samples, miR-885-5p and miR-193a-5p
Stachowiak et al. [167]	EVs in sputum, exhaled breath condensate, and serum	Eight paediatric patients during pulmonary exacerbation and seven stable, aged 6–8 years	miRNA profiling by next-generation sequencing	The four miRNAs with the greatest fold-change between stable and exacerbation in sputum and serum (let-7c, miR-16, miR-25-3p, and miR-146a) were selected for validation. A panel of all four miRNAs in serum was the best predictive model of exacerbation, with miR-146a improving the predictive model of C-reactive protein and neutrophilia. Expression of airway miR-25-3p improved the diagnostic value of FEV1% predicted and FVC% predicted

EVs, extracellular vesicles; FEV1%, forced expiratory volume in 1 s predicted; FVC%, forced vital capacity predicted; HC, healthy controls.

**Table 4 cimb-48-00058-t004:** Cystic fibrosis-associated miRNAs across cellular and extracellular compartments.

miRNAs Found	AECs	Monocytes/Macrophages	Neutrophils	Extracellular Circulating
miR-155-5p	✓		✓	
miR-636	✓		✓	
miR-181a-5p	✓			
miR-126-3p	✓			
miR-17	✓			
miR-199a-3p	✓			
miR-93	✓			
miR-101-3p	✓			
miR-199a-5p		✓		
miR-146a		✓		✓
miR-224-5p		✓		
miR-452-5p		✓		
miR-223-3p	✓		✓	
miR-106a-5p			✓	
miR-221-3p			✓	
miR-370-3p	✓			
miR-886-5p	✓			
miR-10b-3p	✓			
miR-577-5p	✓			
miR-1257	✓			
miR-449	✓			
miR-145-5p	✓			
miR-494	✓			
miR-509-3p				
miR-885-5p				✓
miR-486-5p				✓
miR-193a-5p				✓
let-7b-5p				✓
let-7c-5p				✓
let-7d-5p				✓
miR-103a-3p				✓
miR-598-3p				✓
miR-16				✓
miR-25-3p				✓

## Data Availability

Table 2 is an author-curated summary derived from publicly available GEO DataSets. No original data were generated in this study.

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
