# Peer review of "The Involvement of MicroRNAs in Innate Immunity and Cystic Fibrosis Lung Disease: A Narrative Review"

_cimb, 2026, doi:10.3390/cimb48010058_

Round 1

Reviewer 1 Report

Comments and Suggestions for Authors

Major points:

1) This review manuscript lacks a dedicated methods section. For example, this could comprise "In order to accomplish a deepen insight into the relationship of miRNAs with CF, various genomic and transcriptomic studies have been performed by using immortalized and primary epithelial cells and were selected from the GEO Dataset [112] by using as keywords: “Cystic fibrosis” AND “miRNA Expression” AND “Homo sapiens”. From 67 studies found, three which were complying with AECs were extracted (Table 2)." (line 414).

2) The immunohistochemical panel presented in Figure 1 lacks sufficient resolution. Please increase its size and/or pixel density.

3) The "6. Conclusions and future outlook" bears signs of a discussion section as it contains a plethora of references to previous studies. Please either separate this chapter into two or narrow down its scope to include only the conclusions and the future outlook.

Minor points:

1) Please define abbreviation for "OMIM" (line 37), "IL-13" (line 100), "Th2" (line 100), "TLRs" (line 102), "NOD" in "NOD-like" (line 102), "TNF" (line 107), "RIG-I" (line 121), "MDA5" (line 121), "BECs" (line 123), "poly(IC)" (line 124), "PI3K" (line 173), "NFKB1" (line 178), "RELA" (line 178), "FOS" (line 179), "JUN" (Line 179), "CASP3" (line 179), "SHIP1" (line 199), "RANK" (line 206), "IGF1" (line 213), "WISP1" (line 213), "SMAD2" in "SMAD2-dependent" (line 334), "NETosis" (line 387), "NET" (line 389), "CREB" (line 394), "PDLIM2" (line 409), "SOCS1" (line 409), "NOS-2" (line 444), "GILT" (line 445), "MHC" (line 446).

2) Please change "Transmembrane Conductance Regulator" to "transmembrane conductance regulator" (line 42).

3) Please provide reference for "Opportunistic bacterial pathogens infesting the CF airways are comprised of both Gram-positive (Staphylococcus aureus) and Gram-negative (Haemophilus infleunzae, Burkholderia cepacia, Pseudomonas aeruginosa) species" (line 45).

4) Please provide reference for "Viral infections often superimpose on bacterial infections and are the cause of pulmonary exacerbation during the course of a chronic status" (line 48).

5) Please replace "the associated" with something like "associated with" (line 51).

6) Please change "CF lung disease (CFLD)" to "CFLD" (line 55).

7) Please replace "neutrophils’ infiltration" with "infiltration of neutrophils" (line 56).

8) Please provide reference for "miRNAs are important in regulation of innate immune defense and inflammatory processes, as their dysregulation contribute to chronic inflammatory lung diseases like CF" (line 67).

9) Please change "like" to "such as" (lines 69, 259, 264).

10) Please provide reference for "Airway epithelial cells (AECs) can be considered an important barrier to mucosal penetration of microorganism due to its tightness" (line 81).

11) Please provide reference for "Moreover, in addition to provide an effective mucociliary clearance of the lower airspace, AECs are relevant players in the immune response to endogenous and exogenous molecular patterns" (line 82).

12) Please replace "provide" with "providing" (line 82).

13) Please provide reference for "Type I immunity in AECs is induced in response to pathogen-associated molecular patterns (PAMPs) and damage-associated molecular patterns (DAMPs) through pattern-recognition receptors (PRRs), which activate antimicrobial and pro-inflammatory responses to effect innate and acquired immunity" (line 88).

14) Please change "MUC5AC [20], which traps pathogens in the airway lumen" to "MUC5AC, which traps pathogens in the airway lumen [20]" (line 94).

15) Please provide reference for "AEC type 3 is involved in killing extracellular bacteria, fungi and in inactivating viruses during the extracellular phase of their infectious cycle" (line 100).

16) Please replace "bacteria, fungi" with "bacteria and fungi," (line 101).

17) Please provide reference for "TLRs, NOD-like receptors (NLRs), and complement receptors are involved in sensing PAMPs and DAMPs derived from tissue and cell damage exerted by these pathogens and by the inflammation itself" (line 102).

18) Please provide reference for "Cytokines involved in the amplification of these responses are IL-22 and multiple IL-17 family members, which operate to recruit neutrophils, the effector cell type in this type of immunity" (line 104).

19) Please change "CF" to "cystic fibrosis" (lines 110, 185, 307, 377, 398).

20) Please replace "AMPs’ activities" with either "the activity of AMPs" or "AMP activity" (line 116).

21) Please provide reference for "While infection (1 h) with major group RV16 of primary CF AECs resulted in a trend towards a diminished IFN response at the level of IFN-λ1, IFN-λ2/3 and IFN-β, PRRs TLR3, RIG-I and MDA5 and ISGs in comparison to healthy AECs, the IFN pathway induction upon minor group RV1B infection was significantly increased at the level of IFNs and PRRs in CF BECs compared to healthy BECs" (line 119).

22) Please change "RIG-I and MDA5" to "RIG-I, and MDA5," (line 121).

23) Please replace "poly(IC)" with "poly(I:C)" (line 124).

24) Please change "IL-6" to "IL-6," (lines 125, 132, 283).

25) "Previously, Dauletbaev et al. had shown, upon infection with RV16 for 24 hrs, no differences between primary CF HBE and healthy cells in the secretion of the antiviral cytokine IFN-β and the pro-inflammatory cytokine IL-8 as well as in the up-regulation of the interferon-responsive gene OAS1 [31], highlighting that the experimental conditions (short vs. continuous exposure) may change the magnitude of response by CF cells" (line 125) is way too long. Please split into at least two sentences.

26) Please replace "shown, upon infection with RV16 for 24 hrs, no differences between primary CF HBE and healthy cells in the secretion of the antiviral cytokine IFN-β and the pro-inflammatory cytokine IL-8 as well as in the up-regulation of the interferon-responsive gene OAS1" with "shown no differences between primary CF HBE and healthy cells in the secretion of the antiviral cytokine IFN-β and the pro-inflammatory cytokine IL-8 as well as in the up-regulation of the interferon-responsive gene OAS1 upon infection with RV16 for 24 hr" (line 125).

27) Please replace "up-regulation" with "upregulation" (line 128).

28) Please change "cells" to "cells (AECs)" (line 138).

29) Please replace "TLRs 1–10" with "TLR1–10," (line 146).

30) Please change "epithelia [35-37], thereby allowing CF AECs participating in type 1 and 3 immune modules" to "epithelia, thereby allowing CF AECs participating in type 1 and 3 immune modules [35-37]" (line 146).

31) Please replace "airway epithelial cells" with "AECs" (line 151).

32) Please change "TLRs’ expression" to "the expression of TLRs" (line 154).

33) Please replace "transporters" with "transporters," (line 158).

34) Please change "AECs" to "AEC" (line 166).

35) Please replace "localization [51], which is necessary for AKT phosphorylation" with "localization, which is necessary for AKT phosphorylation [51]" (line 173).

36) Please change "Non-Cystic Fibrosis" to "non-cystic fibrosis" (line 176).

37) Please replace "dysregulations" with "dysregulation" (line 177).

38) Please change "RELA" to "RELA," (line 178).

39) Please replace "MYL12B" with "MYL12B," (line 181).

40) Please change "IRF1, IRF3, IRF5, and IRF7" to "IRF1,3,5, and 7" (line 182).

41) Please replace "phenotype, and" with "phenotype and" (line 182).

42) Please replace "AEC" with "airway epithelial cell" (line 185).

43) Please define abbreviation for "CF", "ALI", "AECs", "SHIP1", "IL-1R1", "IKK-β", "RANK", "IGF1", "WISP1", "TOM1", "Tollip", "ND", "PI3K", "MAPK" in Table 1.

44) Please remove bold formatting from "miR-155" (line 194).

45) Please change "lungs" to "lung" (line 194).

46) Please provide reference for "IB3-1 CF cells presented higher expression levels of miR-155 and let-7c than wild type CFTR-repaired daughter cell, IB3-1/S9, a result that was confirmed ex-vivo in primary bronchial AECs and blood neutrophils only for miR-155" (line 195).

47) Please remove bold formatting from "miR-636" (line 202).

48) Please remove bold formatting from "miR-181a-5p" (line 211).

49) From "A miR-181a-5p inhibitor led to an increase in the mRNA levels of the growth factor IGF1 and WISP1, a key component of cell proliferation/migration programs, confirming that miR-181a-5p may play a role in their deregulation" (line 214) is not unequivocally clear whether the authors mean to say "cell proliferation and migration" or "cell proliferation or migration"? Please fix.

50) Please remove bold formatting from "miR-101-3p" (line 217).

51) Please replace "brought to" with something like "stimulated" (line 218).

52) Please remove bold formatting from "MiR-126" (line 223).

53) Please change "TOM1 (target of Myb1)" to "target of Myb1 (TOM1)" (line 224).

54) Please replace "Tollip (Toll-interacting protein)" with "Toll-interacting protein (Tollip)" (line 224).

55) Please provide reference for "CF bronchial brushings and CF cell lines showed upregulation of TOM1 and Tollip as compared to their non-CF counterparts" (line 226).

56) Please remove bold formatting from "miR-17" (line 234).

57) Please change "brushings, and" to "brushings while" (line 234).

58) Please replace "medium" with "medium," (line 236).

59) Please remove bold formatting from "miR-199a-3p" (line 238).

60) Please provide reference for "In a CF cell line (CFBE41o−) miRNA mimic suppressed NF-κB pathway by directly targeting KBKB, which encodes IKKβ protein, and lowered IL-8 secretion" (line 241).

61) Please remove bold formatting from "miR-93" (line 243).

62) Please provide reference for "The two major macrophage populations of the lung are those located in the airway lumen, i.e. alveolar (AMs), and those residing in the lung parenchyma, i.e. interstitial" (line 249).

63) Please provide reference for "They remove cellular and extracellular matrix debris and recycle surfactant molecules, maintaining immunological and physiological homoeostasis" (line 251).

64) Please change "causing" to something like "triggering" (line 254).

65) Please provide reference for "AMs exert their protective effects via non-specific lines of defence, such as high phagocytic ability, the secretion of antimicrobials, NO, as well as chemokines and cytokines (TNF-α, IL-1β, IFN-γ) that recruit and activate neutrophils, monocytes, and DCs" (line 255).

66) Please replace "defence, such" with "defence such" (line 256).

67) Please change "the secretion" to "secretion" (line 256).

68) Please replace "antimicrobials, NO" with "antimicrobials and NO" (line 256).

69) Please change "IFN-γ" to "and IFN-γ" (line 257).

70) Please replace "IL-1RA" with "IL-1RA," (line 259).

71) From "IMs perform homeostatic and immunoregulatory/inflammatory functions" (line 260) is not unequivocally clear whether the authors mean to say "immunoregulatory and inflammatory functions" or "immunoregulatory or inflammatory functions"? Please correct.

72) Please provide reference for ""IMs perform homeostatic and immunoregulatory/inflammatory functions" (line 260).

73) Please change "tissue, by" to "tissue by" (line 263).

74) Please replace "IGF-1" with "and IGF-1" (line 264).

75) From "M1 macrophages participate into Type 1 immune response, in such they are activated by IFN-γ, lipopolysaccharide and/or TNFα, produce proinflammatory cytokines, are involved in direct destruction of intracellular pathogens, and promote a local Th1 environment" (line 268) is not unequivocally clear whether the authors mean to say "lipopolysaccharide and TNFα" or "lipopolysaccharide or TNFα"? Please fix.

76) Please provide reference for "M1 macrophages participate into Type 1 immune response, in such they are activated by IFN-γ, lipopolysaccharide and/or TNFα, produce proinflammatory cytokines, are involved in direct destruction of intracellular pathogens, and promote a local Th1 environment" (line 268).

77) Please change "into Type" to "in type" (line 268).

78) Please replace "in such" with "such that" (line 268).

79) Please change "lipopolysaccharide" to "lipopolysaccharide," (line 269).

80) Please provide reference for "M2 macrophages are characterized by their participation in Type 2 immune responses and thus arise in response to IL-4 and IL-10" (line 271).

81) Please replace "Type" with "type" (line 271).

82) Please provide reference for "M2 macrophages subsets exert different functions: M2a facilitate parasite encapsulation and destruction, M2b are immunoregulatory, and M2c are involved in tissue remodelling and matrix deposition" (line 272).

83) Please change "functions:" to "functions." (line 273).

84) Please replace "that, in" with "that in" (line 275).

85) Please change "It is likely that, in front of this simplistic view, pulmonary macrophages, extremely plastic cell, tailor" to something like "Owing to their extremely plastic architecture, pulmonary macrophages tailor" (line 275).

86) Please change "signalling" to "signaling" (lines 280, 285).

87) Please replace "In inflammatory conditions, besides tissue resident macrophages, neutrophils and monocytes are attracted into the lung" to something like "Besides tissue-resident macrophages, neutrophils and monocytes are attracted into the lung under inflammatory conditions" (line 289).

88) Please change "progression" to "progression," (line 291).

89) Please replace "12-hydroxyeicosatetraenoic" with "and 12-hydroxyeicosatetraenoic" (line 293).

90) Please change "lung, in" to "lung in" (line 295).

91) Please replace "(CCR2) dependent" with "(CCR2)-dependent" (line 296).

92) Pleae change "(CCL2; MCP-1)" to something like "(CCL2) (also known as MCP-1)" (line 297).

93) Please provide reference for "These recruited macrophages can either acquire a AM-like phenotype or are phenotypically distinct, providing functional roles in epithelial repair, induction of fibrosis, or enhanced inflammation" (line 297).

94) Please replace "a" with "an" (line 298).

95) Please change "expands, and" to "expands and" (line 301).

96) Please provide reference for "CF lungs host tissue resident and recruited macrophages, these last of circulating monocyte origin" (line 309).

97) "this" appears twice in "Based on this, this section will address the both of these population" (line 310). Please rephrase.

98) Please replace "these" with "which are the" (line 309).

99) Please provide reference for "Based on this, this section will address the both of these population" (line 310).

100) Please change "CF macrophages present above all" to either "Above all, CF macrophages present" or "CF macrophages present" (line 312).

101) Please replace "pathogens, such" with "pathogens such" (line 313).

102) Please change "Reduced expression of CD11b, in the context of complement-mediated opsonisation [88], and" to "In the context of complement-mediated opsonisation [88], reduced expression of CD11b and" (line 314).

103) Please replace "insulin-like growth factor 1" with "IGF1" (line 317).

104) Please change "macrophages, derived" to "macrophages derived" (line 320).

105) Please replace "TNF-α" with "and TNF-α" (line 321).

106) Please change "heightened, and" to "heightened and" (line 321).

107) Please replace "sorely" with something like "profoundly" (line 322).

108) Please change "to, and" to "to and" (line 326).

109) Please replace "[94], and" with "[94] and" (line 326).

110) Please provide reference for "Upon lung damage, classic Ly6ChiCCR2+ monocytes (cMons) are recruited from the circulation in a C-C motif chemokine receptor 2 (CCR2) dependent manner" (line 328).

111) Please change "C-C motif chemokine receptor 2 (CCR2) dependent" to "CCR2-dependent" (line 329).

112) Please provide reference for "CCR2+ monocyte recruitment and monocyte-derived macrophages were increased in CF mouse lungs in response to chronic LPS, which would sustain a pro-inflammatory environment by facilitating neutrophil recruitment" (line 331).

113) Please replace "contributed" with "contribute" (line 334).

114) Please provide reference for "Macrophages are involved not only in the initiation (M1) but also in the resolution of lung inflammation (M2)" (line 337).

115) Please change "with" to "together with" (line 341).

116) Please replace "reduced of" with "reduced" (line 342).

117) Please change "macrophages, could" to "macrophages could" (line 342).

118) Please replace "not" with "no" (line 343).

119) Please change "CF lung disease" to "CFLD" (lines 346, 434, 491).

120) Please remove bold formatting from "miR-199a-5p" (line 350).

121) Please replace "(derived from murine bone marrow and human peripheral blood)" with "derived from murine bone marrow and human peripheral blood" (line 350).

122) Please provide reference for "By targeting CAV1, its elevated expression promotes CF-associated lung hyper-inflammation through disruption of CAV1-mediated inhibition of TLR4 signaling" (line 351).

123) Please change "its elevated expression" to "the elevated expression of miR-199a-5p" (line 352).

124) Please provide reference for "This axis is regulated by PI3K/AKT signaling, which is blunted in CF macrophages, leading to increased level of miR-354 199a-5p and decreased levels of CAV1" (line 353).

125) Please provide reference for "In summary, CF macrophages have a blunted PI3K/pAKT signaling in response to TLR4 activation, which leads to accumulation of miR-199a-5p, decreased CAV1 expression, ultimately leading to ineffective negative feedback of TLR4 signaling and hyper-inflammation" (line 355).

126) Please provide reference for "Following miRNA profiling in peripheral blood monocytes-derived macrophages from CF and non-CF individuals, a panel of differentially expressed miRNAs in CF macrophages, as compared to non-CF macrophages, was identified" (line 359).

127) Please remove bold formatting from "miR-146a" (line 361).

128) Please change "up-regulated" to "upregulated" (lines 363, 448).

129) Please provide reference for "Moreover, inhibition of miR-146a reduces IL-6 production in LPS-stimulated CF macrophages" (line 365).

130) Please remove bold formatting from "miR-224-5p" (line 369).

131) Please replace "(3.0-fold; P = 0.1336)" with "(3.0-fold) (P = 0.1336)" (line 370).

132) Please remove bold formatting from "miR-452-5p" (line 370).

133) Please change "(1.9-fold; P = 0.1242)" to "(1.9-fold) (P = 0.1242)" (line 370).

134) Please provide reference for "SMAD4 and miR-224-5p levels were negatively correlated in CF monocytes, although not significantly" (line 372).

135) Please replace "monocytes [101], a mechanism to control hyper-inflammation" with "monocytes, a mechanism to control hyper-inflammation [101]" (line 374).

136) Please provide reference for "As recalled above, neutrophils are the first line of innate immune cells occurring during sterile and non-sterile lung inflammatory conditions and recruited from the blood" (line 379).

137) Please change "abnormalities, such" to "abnormalities such" (line 381).

138) Please remove "As to neutrophils’ survival and function once they have entered into the CF lungs, there are variable observations that jeopardize the issue" (line 384).

139) "Some studies had reported that neutrophils die prematurely upon recruitment to CF airways, via either secondary necrosis following apoptosis and lack of macrophage efferocytosis, or NETosis [105,106], while others show that CF neutrophils have a primary defect causing decreased spontaneous apoptosis and allowing increased levels of NET formation that can promote inflammation" (line 385) is too long. Please split into two sentences.

140) Please replace "airways, via" with "airways via" (line 386).

141) Please change "or" to "or via" (line 387).

142) Please replace "(termed “GRIM”)" with "termed "GRIM"" (line 392).

143) Please provide reference for "GRIM neutrophils undergo active exocytosis of primary granules, leading to massive release of elastase and myeloperoxidase that damage the airway tissue and perpetuate inflammation; upregulate CREB and mTOR prosurvival pathways; and show augmented cell surface nutrient transporter expression and glucose uptake" (line 392).

144) Please change "inflammation;" to "inflammation," (line 394).

145) Pleae replace "pathways;" with "pathways," (line 395).

146) Please change "upregulated, but" to "upregulated but" (line 399).

147) Please replace "RANK, as" with "RANK as" (line 401).

148) Please provide reference for "Moreover, a significant positive correlation was observed between miR-221-3p and IL-1β transcript, a finding consistent with the proinflammatory role of this microRNA" (line 409).

149) Please replace "microRNA" with "miRNA" (lines 411, 520).

150) Please change "deepen" to "deeper" (line 414).

151) Please define abbreviation for "CF", "non-CF", "AECs", "ALI" in "ALI-polyps", "NCF", "CFTR" in "F508del-CFTR", "TOM1", "IL-11", "IL6R", "WISP1", "isomiRNAs" in Table 2.

152) Please replace "Immortalized CF AECs:" with "Immortalized CF AECs," (2x), "Immortalized non-CF AECs:" with "immortalized non-CF AECs," (2x), "(CF vs NCF)" with "(CF vs NCF);" (3x), "Bronchial cells" with "bronchial cells", "controls:" with "controls,", "acted as" with "acting as a", "IL-18 genes" with "IL-18 genes,", "up-regulated" with "upregulated", "miRNAs expression" with "miRNA expression", "like" with "such as", "airway epithelial cells" with "AECs", "CFTR- UTR3’" with "the CFTR-UTR3’", "of CFTR" with "of the CFTR", "ALI (air liquid interface)" with "air-liquid interface (ALI)", "cystic fibrosis" with "CF", "that contribute" with "that contributes", "proposing new" with "proposing a new", "impairs the CFTR" with "impairs CFTR", "problem of CFTR activity" with "CFTR" in Table 2.

153) Please change "causing" to something like "resulting in", "referred to" to something like "ascribed to the" in Table 2.

154) Please remove italics formatting from "F508del", "CFTR" in Table 2.

155) Please replace "Based in" with "Based on" (line 423).

156) Please provide reference for "miR125-3p.1 has a target IL6ST (interleukin 6 cytokine family signal transducer), the subunit gp 130 of the receptor for IL-6" (line 427).

157) Please change "IL6ST (interleukin 6 cytokine family signal transducer)" to "interleukin 6 cytokine family signal transducer (IL6ST)" (line 427).

158) Please replace "gp 130" with "gp130" (line 428).

159) Please provide reference for "Interestingly, it has been recently shown that a CF bronchial cell line is hyper-responsive to both IL-6 classic and trans-signaling as compared to non-CF counterparts" (line 428).

160) Please change "levels" to "level" (line 430).

161) Please provide reference for "However, IL-6 is also anti-inflammatory" (line 431).

162) From "Since Oglesby et al. [57] already showed that CF cells show an increase of TOM1 (target of Myb 1), a negative regulator of the NF-κB pathway, in consequence of miR-126 down-regulation, possibly exerting a compensation for the hyper-inflammation occurring in CF lung disease, the positive regulation of IL6ST is in line with the inflammation attenuation" (line 431) is not exactly clear what the authors mean by "increase of TOM1"? Are they referring to TOM1 expression?

163) "Since Oglesby et al. [57] already showed that CF cells show an increase of TOM1 (target of Myb 1), a negative regulator of the NF-κB pathway, in consequence of miR-126 down-regulation, possibly exerting a compensation for the hyper-inflammation occurring in CF lung disease, the positive regulation of IL6ST is in line with the inflammation attenuation" (line 431) is way too long and complex. Please split into two sentences.

164) Please replace "TOM1 (target of Myb 1)" with "TOM1" (line 432).

165) Please change "down-regulation" to "downregulation" (line 433).

166) From "miR-223-3p and miR-145-5p are upregulated in CF AECs and cause not only suppression of CFTR mRNA/protein level [114], but also regulate inflammation and immune responses" (line 436) is not unequivocally clear whether the authors mean to say "CFTR mRNA and protein level" or "CFTR mRNA or protein level"? Please correct.

167) Please change "level [114], but also regulate inflammation and immune responses" to "level but also regulate inflammation and immune responses [114]" (line 437).

168) Please provide reference for "miR-223-3p has a target IL6ST that should be then downregulated, witnessing a complex regulation of inflammatory signals between AECs and neutrophils" (line 438).

169) "On the other hand, the finding that miR-223-3p has as targets IFIH1 (alias MDA-5) and CLEC14A (C-type lectin domain family 14, member A), testifying that defensive immune responses to virus (MDA-5), fungi (CLEC14A), are altered and likely not properly functioning" (line 439) is too complex. Please simplify its structure.

170) Please provide reference for "On the other hand, the finding that miR-223-3p has as targets IFIH1 (alias MDA-5) and CLEC14A (C-type lectin domain family 14, member A), testifying that defensive immune responses to virus (MDA-5), fungi (CLEC14A), are altered and likely not properly functioning" (line 439).

171) Please replace "CLEC14A (C-type lectin domain family 14, member A)" with "C-type lectin domain family 14, member A (CLEC14A)" (line 440).

172) Please change "(MDA-5), fungi" to "(MDA-5) and fungi" (line 442).

173) "Similarly, miR-145-5p has as targets IFNGR2 (interferon gamma receptor 2) that controls downstream NOS-2, whose reduced expression would hamper the AEC response to pathogens [50], and IFI30 (a lysosomal thiol reductase induced by IFN-γ, also known as GILT), processing peptides for MHC class II binding and presentation, thus inhibiting antigen recognition by T cells" (line 442) is too long and complex. Please simplify its structure by splitting into at least two sentences.

174) Please provide reference for "and IFI30 (a lysosomal thiol reductase induced by IFN-γ, also known as GILT), processing peptides for MHC class II binding and presentation, thus inhibiting antigen recognition by T cells" mentioned in "Similarly, miR-145-5p has as targets IFNGR2 (interferon gamma receptor 2) that controls downstream NOS-2, whose reduced expression would hamper the AEC response to pathogens [50], and IFI30 (a lysosomal thiol reductase induced by IFN-γ, also known as GILT), processing peptides for MHC class II binding and presentation, thus inhibiting antigen recognition by T cells" (line 442).

175) Please replace "IFNGR2 (interferon gamma receptor 2)" with "interferon gamma receptor 2 (IFNGR2)" (line 443).

176) Please change "IFI30 (a lysosomal thiol reductase induced by IFN-γ, also known as GILT)" to "IFI30, a lysosomal thiol reductase induced by IFN-γ, also known as GILT" (line 445).

177) Please replace "miR-577-5p" with "and miR-577-5p" (line 449).

178) Please provide reference for "BACH1 (BTB domain and CNC homolog 1), a target of miR-155-5p, is a transcription factor and a master regulator of antioxidant systems; it controls also the expression of CFTR in airway epithelium by either direct occupancy of cis-regulatory elements (CREs) or modulating expression under oxidative stress" (line 451).

179) Please change "BACH1 (BTB domain and CNC homolog 1)" to "BTB domain and CNC homolog 1 (BACH1)" (line 451).

180) Please replace "systems; it" with "systems. It" (line 452).

181) Please change "HO-1 (Heme Oxygenase 1)" to "heme oxygenase 1 (HO-1)" (line 457).

182) Please replace "Also Bhattacharyya" with "Bhattacharyya" (line 459).

183) Please change "over expressed" to "overexpressed" (line 459).

184) Please replace "Its" with "miR-155-5p" (line 460).

185) Please change "SHIP1 (phosphatidylinositol-3,4,5-trisphosphate 5-phosphatase 1)" to "phosphatidylinositol-3,4,5-trisphosphate 5-phosphatase 1 (SHIP1)" (line 461).

186) Please replace "enhancing the" with "enhancing" (line 462).

187) Please provide reference for "Activation of PI3K/AKT signaling system attracts immune cells likely to enhance inflammation with upregulated expression of the IL-8 chemokine" (line 462).

188) Please change "nasal and" to "nasal, and" (line 466).

189) Please replace "polyps" with "polyp" (line 466).

190) Please change "WISP1 (WNT1 inducible signaling pathway)" to "WNT1 inducible signaling pathway (WISP1)" (line 468).

191) Please provide reference for "These microRNAs are predicted to target key regulator of inflammation, such as IGF1 (insulin-like growth factor 1)" (line 470).

192) Please replace "microRNAs" with "miRNAs" (lines 470).

193) Please change "inflammation, such" to "inflammation such" (line 471).

194) Please replace "IGF1 (insulin-like growth factor 1)" with "insulin-like growth factor 1 (IGF1)" (line 471).

195) "While direct evidence for IGF1 as a downstream target in this axis within CF lung tissue remains limited, the broader literature demonstrates that patients with cystic fibrosis exhibit both systemic and local deficiency of IGF‑1, which is associated with impaired alveolar macrophage function, particularly reduced bactericidal activity against Pseudomonas aeruginosa; this finding suggests that compromised innate immunity in CF lungs is not solely due to epithelial defects but also involves immune cell dysfunction driven by low IGF‑1" (line 472) is way too long and way too complex. Please split into at least three sentences.

196) Please provide reference for "While direct evidence for IGF1 as a downstream target in this axis within CF lung tissue remains limited, the broader literature demonstrates that patients with cystic fibrosis exhibit both systemic and local deficiency of IGF‑1, which is associated with impaired alveolar macrophage function, particularly reduced bactericidal activity against Pseudomonas aeruginosa; this finding suggests that compromised innate immunity in CF lungs is not solely due to epithelial defects but also involves immune cell dysfunction driven by low IGF‑1" (line 472).

197) From "While direct evidence for IGF1 as a downstream target in this axis within CF lung tissue remains limited, the broader literature demonstrates that patients with cystic fibrosis exhibit both systemic and local deficiency of IGF‑1, which is associated with impaired alveolar macrophage function, particularly reduced bactericidal activity against Pseudomonas aeruginosa; this finding suggests that compromised innate immunity in CF lungs is not solely due to epithelial defects but also involves immune cell dysfunction driven by low IGF‑1" (line 472) is not unequivocally clear what the authors mean by "low IGF‑1"? Are they referring to IGF-1 expression?

198) Please change "function, particularly" to "function and particularly" (line 475).

199) Please replace "aeruginosa; this" with "aeruginosa. This" (line 476).

200) Please change "(Impaired inflammaotry signalling)" to "impaired inflammatory signalling", "(Impaired response to virus)" to "impaired response to virus" or "impaired viral response", "(Impaired response to fungi)" to "impaired response to fungi" or "impaired fungal response", "(Impaired antigen presentation)" to "impaired antigen presentation", "(Wound healing and lung inflammation)" to  "wound healing and lung inflammation", "(compensation for hyper-inflammation)" to "compensation for hyper-inflammation", "(Impaired response to pathogens)" to "impaired response to pathogens" or "impaired pathogen response", "(decreased hypoxia injury)" to "decreased hypoxia injury", "(Increased inflammation)" to "increased inflammation" in Figure 1.

201) miRNAs depicted in the blue circles in Figure 1 seem to suffer from sufficient lack of space. Please increase circle radius for each miRNA.

202) "IFI30" seems not to be vertically well aligned with other genes in the yellow boxes in Figure 1. Please correct.

203) It might be worth presenting the immunohistochemical panel of Figure 1 either above or below the neighboring signal transduction map.

204) Please change "Fig." to "Figure" (line 482).

205) Please replace "arrow:" with "arrow," (line 483 2x).

206) Please change "downregulation. Thick" to "downregulation; thick" (line 483).

207) Please replace "upregulation. Thin arrows:" with "upregulation; thin arrows," (line 483).

208) Please change "Human trachea, with" to "human trachea with" (line 485).

209) Please replace "cells (C), goblet cells (G)" with "(C), goblet (G)," (line 485).

210) The link "https://commons.wikimedia.org/wiki/File:Masson%27s_trichrome_staining_of_the_Human_trachea.jpg)." seems to be dysfunctional. Please fix.

211) Please change "tissues" to "tissues," (line 503).

212) Please replace "give a deepen" with "provide a deeper" (line 503).

213) Please provide reference for "For example, as a start to understand whether extracellular miRNAs can potentially identify a pulmonary exacerbation (PE) in CF, EV-derived miRNA in sputum, exhaled breath condensate, and serum were studies in pediatric patients during PE and the stable stage of CF" (line 511).

214) Please change "In" to "Along" (line 519).

215) Please replace "neutrophils" with "and neutrophils" (line 525).

216) Please change "A.C." to "A.C.," (line 531).

Reviewer 2 Report

Comments and Suggestions for Authors

Thank you for the kind invitation to review the current manuscript, which offers an updated overview of the role of each immune cell, described and interpreted in the light of miRNA expression and function. Also address the participation in inflammation and wound healing. 

I have some comments for the authors to address.

Please include "a narrative review" at the end of the title, since this corresponds to this type of paper.

In the abstract, please structure it by sections and in the paragraph "The studies we identified foresee the involvement of miRNAs in different processes associated with CFLD, namely impaired response to various classes of pathogens, compensation for hyper-inflammation, altered antigen presentation, and wound healing, at the levels of AECs and macrophages." is a little wordy; I suggest replacing it with: 
The studies we identified suggest that miRNAs are involved in various processes related to CFLD, including impaired pathogen response, compensation for hyper-inflammation, altered antigen presentation, and wound healing, at the levels of AECs and macrophages.

The tables are an effective tool for summarizing information. Please ensure the information is not repeated in the text, particularly in Table 2.

Figure 1 needs careful tuning; note that some text needs adjustment, and consider replacing it with a more prominent style.

Finally, please review the references. Many review papers have been used for the current review; please refer to the original paper describing the initial finding. Also, ensure that you are using either American or British English, but not a mix of both.

Comments on the Quality of English Language

Some paragraphs need revision. As I mentioned earlier, they should ensure they are using either American or British English, but not both.

Reviewer 3 Report

Comments and Suggestions for Authors

Carbone and colleagues provide a review on the role of micro-RNAs for lung inflammation and innate immunity in cystic fibrosis.

This is a highly timely topic for which the authors have consulted 111 references. The manuscript is well written and can be clearly understood, however, the following sections lack focus on the topic chosen by the authours for their review:

Major:

1. Focus of manuscript. Chapter 2 describe the immunologicalproperties of AEC in detail - but between line 79 and line 183, miRNA do not play a role. The following Paragraphs 2.2.1 and 2.2.2  are focussing on miRNA. A similar structure is seen for the subchapter on macrophages, where again the authors again firstly review macrophage function (no miRNA) and their dysfunction in CF (no miRNA), to then provide details on the miRNA implicated in miRNA regulation. 
1a. While this structure is understandable, a modification of either the topic and title or a better link between the "introductory" subchapters and the miRNA-subchapters is advisable to sharpen the review's focus. 
1b. The observation that "protein X is involved in the function of a cell A in a CF-relevant manner" and at the same time "miRNA Y has an influence on transcript X" makes it plausible, albeit not a proof that "miRNA Y regulates protein X in cell A" as miRNA can be secreted and signal to other cells nearby, for instance in extracellular vesicles. This would be one argument in favor of merging the subchapters to provide a clearer focus on how miRNA take part in CF lung disease: if miRNA are found in a cell line like CFBE41o-, no other cell type than a basic AEC is around (albeit, unless grown at air-liquid-interface, the polarity of CFBE40o- and thus their AEC characteristics are limited). But in contrast, a nasal sample can contain immune cells as well as epithelial cells, and it depends on how the biomaterial is processed which cells contribute to the miRNA observed. 
1c. In the line with the last argument, please describe the biomaterials under study for miRNA-relevant findings in more detail, making it feasible to understand whether the source and the point of action of the miRNA is the same - or alternatively, if a CFTR-deficient AEC "might use" miRNA species to signal towards the innate immune cell compartment. While the latter discussion might extend the scope of the present review, at least the biomaterials studied within the references should be described in sufficient detail in text and table to indicate the likely source of the miRNA, describe whether the cell type is uniform or mixed, and provide a brief insight whether the signalling is "within one cell type" (as currently inferred by the subchapter structure) or can be "between cells".

2. Which particular AEC cell type are adressed by the authours? In the aera of scRNASeq, it should be feasible to account for the cellular diversity of AECs.

3. Without a doubt, macrophages from CF patients behave differently when compared to macrophages from healthy individuals. But: the "CFTR deficiency in human macrophages" is just one out of two hypotheses explaining why CF-derived macrophages behave differnetly, the other being that epigenetic or regulatory changes in CF-derived macrophages make them irreversibly different, likely due to the altered inflammatory state of CF individuals and the close proximity of resident macrophages to truely CFTR-defincient AEC. The literature is unclear on which hypothesis is correct, albeit, the proof that macrophages express CFTR protein is not convincing in the literature, whereby as many "yes they do" as "no they don't" publications can be found. As this particular controversy is not the topic of the review, kindly rephrase the "CFTR deficiency in human macrophages" to a more neutral description of aberrant macrophage behavious in CF.

4.Figure 1: 
4a. according to the text in subchapter 5, Figure 1 relies on the transcriptome findings as shown in Table 2. Biomaterials therein are primray cells, but also cell lines and thus, the displayed pseudostratified epithelium in Figure 1A is misleading. 
4b. OMICs technologies typically provide more than a few target miRNA, and cut-off values for test statistics between articles might differ. Thus, the data compiled in Figure 1A represents a subset of miRNA findings from the OMICs data collection. Please state criteria for inclusion / exclusion of OMICs-generated miRNA findings to make the content of Figure 1 more neutral. Preferentially, OMICs readout between different studies is adjusted to the same criteria for listing  only one out of 93 differentially expressed miRNA (Ref. 57). Why these particular 6 miRNA are selected to be displayed in Figure 1 is unclear.
4c. Reciprocally, the targets for miRNA are typically diverse, albeit Figure 1 shows a biased selection. A network analysis using appropriate tools (miRNet, miRTargetLink, miRWalk, or an adaptation of other miR network analysis examples) would be advantageous to understand how OMICs-derived miR interact and which targets they reach.

5. In their outlook chapter, authors list plasma miRNA as another miRNA ophenotype, studied between CF and non-CF. This is highly relevant to the topic of the review and deserves a further subchapter, similarly to AEC and M. or N. already summarized. Likewise, a table displaying those EC miRNA with putative CF relevance would be helpful.

6. Do the same miRNA which are CF-specifically / CF-typically / with relevance to CF appear in the context of AEC, M., N. or EC?

Minor: 

1. The title is highly engaging, chellenging "a dangerous liaison" between micro-RNAs and CF lung disease. However, the reviwer is convinced that a more neutral title would be more appropriate as MDPI is not "yellow press", but "scinetific communication".

2. Authors have choosen the abbreviation CFLD fpor CF lung disease. This is unfortunate, as CFLD is mostly used to say "CF liver disease". Instead of an abbreviation, the reviewer suggest to write CF lung disease - or, even better, to specifically name the specific subphenotype of CF lung disease as appropriate in the respective context.

3. In line 61, MicroRNA are defined to be abbreviated as miRNA, which is fine. But at least in the title, MicroRN A is written as micro-RNA. Please choose one full-length term (with or wihtout "-") consitently for the entire manuscript. Similarly, choose between MiRNA (caps, see line 63) at the start of a sentence or miRNA at the start of a sentence (no caps, see line 67).

4. Line 74 ff: It is unusual to call epithelial cells "past of the innate immunity" (even though the reviewer agrees). Please distinguish immune cells & non-immune cells such as EC by rephrasing the passage line 74 - line 75, avoiding to subsummarise EC as immune cells 

Round 2

Reviewer 1 Report

Comments and Suggestions for Authors

Major point:

There seems to be way too many small issues (see the minor point list below).

Minor points:

1) "and role of microRNAs" seems to be rather loose in "Involvement of innate immune cells in the pathogenesis of 2 cystic fibrosis lung disease and role of microRNAs: a narrative 3 review 4" (line 2). The role of microRNAs in what?

2) Please change "and" to "and the" (lines 3, 100, 440).

3) "Pulmonary involvement in cystic fibrosis (CF) is a complex disease in which, following respiratory infections caused by bacteria, viruses, and fungi, the inflammatory and immune responses become dysregulated and cause more harm than benefit" (line 16) does not seem to be 100% correct with respect to "Pulmonary involvement in cystic fibrosis (CF) is a complex disease" as one may question why "Pulmonary involvement" is highlighted as "a complex disease", while "cystic fibrosis (CF)" is not.

4) "caused"/"cause" appears twice in "Pulmonary involvement in cystic fibrosis (CF) is a complex disease in which, following respiratory infections caused by bacteria, viruses, and fungi, the inflammatory and immune responses become dysregulated and cause more harm than benefit" (line 16). Please fix.

5) Please replace "disease in which, following respiratory infections caused by bacteria, viruses, and fungi, the inflammatory and immune responses" with "disease, in which inflammatory and immune responses to respiratory infections caused by bacteria, viruses, and fungi" (line 16).

6) Please change "The innate immune response, although essential for the host’s initial defence against these microorganisms, is" to "Although essential for the host’s initial defence against these microorganisms, the innate immune response is" (line 18). 

7) Please replace "cellular components" with "cellular" (line 20).

8) Please change "[AECs]" to "(AECs)" (line 20).

9) Please change "monocytes–macrophages" to "monocytes, macrophages" (lines 20, 665).

10) Please replace "components (cytokines, chemokines, signal transduction pathways, transcription factors)" with "components (cytokines, chemokines, signal transduction pathways, transcription factors)" (line 21).

11) Please change "AECs, as" to "AECs as" (line 23).

12) Please replace "monocytes–macrophages" with "monocytes, macrophages," (line 24).

13) Please change "CF lung disease" to "CF lung disease (CFLD)" (lines 25, 40) and "CF lung disease" to "CFLD" (lines 27, 56, 75, 77, 398, 493, 611, 656, 662).

14) Please replace "hyper-inflammation" with "hyperinflammation" (lines 28, 405, 410, 426).

15) Please replace "healing, at the levels of" with "healing in" (line 29).

16) Please change "it is becoming clear that clinical" to "clinical" (line 29).

17) Please replace "inflammation, not necessarily in this order, since the primary role of these two conditions is being still discussed" with "inflammation" (line 40).

18) "Dysfunction in or lack of CF transmembrane conductance regulator (CFTR), a chloride and bicarbonate channel, leads to abnormal hydration and viscoelasticity of airway mucus, resulting in mucus obstruction of the airways, and thus creating an environment that promotes infection and chronic inflammation, fueling a sort of “vicious cycle”" (line 42) is way too complex. Please simplify the text and/or split into two independent sentences.

19) Would the authors please replace "sort of “vicious cycle”" in "Dysfunction in or lack of CF transmembrane conductance regulator (CFTR), a chloride and bicarbonate channel, leads to abnormal hydration and viscoelasticity of airway mucus, resulting in mucus obstruction of the airways, and thus creating an environment that promotes infection and chronic inflammation, fueling a sort of “vicious cycle”" using a more formal expression (line 42)?

20) Please change "in or" to "of or the" (line 42).

21) Please replace "mucus, resulting" with "mucus resulting" (line 44).

22) Please change "and thus creating" to "creating" (line 44).

23) Please replace "fueling" with "fuelling" (line 45).

24) Please replace "status" with something like "disease" or "malfunction" (line 50).

25) Would the authors please replace "correct repair" in "The wound healing process occurring during damage and repair of the CF airway epithelium is intrinsically altered, however recurrent infections the associated with inflammation in CF airways lead to a cycle of damage and repair of the epithelium surface, not allowing a correct repair" using a more formal expression (line 50)?

26) Please replace "causes" with something like "stimulate" (line 58).

27) Please change "chemokine, interleukin 8 (also known as CXCL8), in" to "chemokine interleukin 8 (also known as CXCL8) in" (line 59).

28) Please replace "patients, causes" with something like "patients leads to the" (line 60).

29) Please provide reference for "MicroRNAs (miRNAs) are small non-coding RNAs that regulate gene expression at the post-transcriptional level, specifically binding with their target mRNAs and inhibiting their translation or causing destruction of them" (line 62).

30) Please change "mRNAs and" to "mRNAs," (line 63).

31) Please replace "causing destruction of them" with something like "inducing their degradation" (line 64).

32) Please change "by the" to "by" (line 66).

33) Please replace "their" with "miRNA" (line 66).

34) Please change "miRNA" to "miRNAs" (line 67).

35) Please replace "and others on" with "and" (line 68).

36) Please change "contribute" to "contributes" (line 70).

37) Please replace "As we shall see, altered" with "Altered" (line 70).

38) Please change "cell types belonging to innate immunity (monocytes and macrophages) and non-immune cells (epithelial cells)" to something like "innate immunity cell types such as monocytes and macrophages and non-immune cells such as epithelial cells" (line 73).

39) Please replace "Due to these multiple" with something like "Given the complexity of" (line 76).

40) Please change "will be described and interpreted in the light of miRNA expression and function" to "in miRNA expression and function will be described and interpreted" (line 77).

41) Please provide reference and/or hyperlink to the "PubMed/MEDLINE and Scopus" databases mentioned in "We conducted a narrative review with a structured search of PubMed/MEDLINE and Scopus using controlled vocabulary and free-text terms for “cystic fibrosis”, “airway epithelial cells”, “macrophages”, “neutrophils”, “innate immunity”, and “microRNAs”" (line 82).

42) Please replace "various genomic" with "genomic" (line 85).

43) Please define abbreviation for "MEDLINE" in "PubMed/MEDLINE" (line 82), "GEO" (line 87), "AI" in "AI-driven" (line 89), "MUC5AC" (line 108), "LYSMD3" (line 111), "IP-10" in "CXCL10/IP-10" (line 143), "HBE" (line 144), "OAS1" (line 146), "IRF5" (line 155), "MyD88" (line 167), "AKT" (line 192), "IKKβ" (line 281), "DCs" (line 304), "IMs" (line 306), "CD11b" (line 364), "sCD14" (line 371), "RANK" (line 456).

44) Please change "Dataset" to "DataSet" (lines 87, 471).

45) Please replace "as" with "the" (line 87).

46) Please change "these responses" to "each response" (line 100).

47) Please replace "and type" with "and" (line 101).

48) Please change "airways, essentially" to "airways" (line 107).

49) Please replace "components, such" with "components such" (line 112).

50) Please change "mannoproteins, via" to "mannoproteins via" (line 113).

51) Please provide reference for "Mucus hypersecretion and chitinase production by AECs are upregulated mostly by interleukin (IL)-13, a T helper (Th) 2 cytokine' (line 113).

52) Please replace "and tumor necrosis factor (TNF)" with "and the tumor necrosis factor α (TNF-α)" (line 123).

53) Please change "the induction of antimicrobial peptides (AMPs) are" to "antimicrobial peptides (AMPs) is" (line 124).

54) Please replace "It has been proposed that AEC" with "AEC" (line 128).

55) Please change "i.e. in" to "in" (line 128).

56) Please replace "lead" with "leads" (line 129).

57) Please change "CFTR dysfunction has as a primary consequence, i.e." to something like "As a primary consequence, CFTR dysfunction results in" (line 130).

58) Please replace "AMP activity" with "activity of AMPs" (line 132).

59) Please change "AECs" to "AEC" (lines 133, 187).

60) Please replace "bronchial epithelial cells" with "bronchial epithelial cells (BECs)" (line 134) and "bronchial epithelial cells" to "BECs" (lines 270, 277, 287, 636).

61) "While infection (1 h) with major group RV16 of primary CF AECs resulted in a trend towards a diminished IFN response at the level of IFN-λ1, IFN-λ2/3 and IFN-β, PRRs TLR3, retinoic acid-inducible gene I (RIG-I), and melanoma differentiation-associated protein 5 (MDA5) and ISGs in comparison to healthy AECs, the IFN pathway induction upon minor group RV1B infection was significantly increased at the level of IFNs and PRRs in CF bronchial epithelial cells (BECs) compared to healthy BECs" (line 135) is way too long. Please split into at least two sentences.

62) It is not entirely clear what the authors mean by the "minor group" in "While infection (1 h) with major group RV16 of primary CF AECs resulted in a trend towards a diminished IFN response at the level of IFN-λ1, IFN-λ2/3 and IFN-β, PRRs TLR3, retinoic acid-inducible gene I (RIG-I), and melanoma differentiation-associated protein 5 (MDA5) and ISGs in comparison to healthy AECs, the IFN pathway induction upon minor group RV1B infection was significantly increased at the level of IFNs and PRRs in CF bronchial epithelial cells (BECs) compared to healthy BECs" (line 135)? Minor in terms of what?

63) Please replace "h" with "hr" (line 135).

64) Please change "PRRs" to "the PRRs" (line 137).

65) Please replace "and" with "as well as" (line 138).

66) Please change "CF bronchial epithelial cells (BECs)" to "CF" (line 140).

67) Please provide reference for "Interestingly, stimulation by polyinosinic:polycytidylic acid (poly(I:C)) – a strong IFN inducer – determined heightened levels of CXCL8/IL-8, IL-6, and CXCL10/IP-10" (line 141).

68) Please replace "CXCL10/IP-10" with "CXCL10/IP-10, a CXCR3 chemokine critically important in the development of a Th1 response to extracellular pathogens" (line 143) and "(IP-10), a CXCR3 chemokine critically important in the development of a Th1 response to extracellular pathogens, and" to "(IP-10), and the" (line 164).

69) Please replace "had" with "have" (lines 143, 438).

70) Please change "HBE and healthy cells" to "and healthy HBEs" (line 144).

71) Please replace "upregulation" with "expression" or "expression level" (line 145).

72) Please change "that the" to "that" (line 149).

73) Please replace "response" with something like "response elicited" (line 150).

74) Please provide reference for "influenza A virus (IAV) infection of CFTR knock-down (KD) AECs showed robust pro-inflammatory response, as evidenced by the IL-8, IL-6, and IP10 gene transcripts" mentioned in "In line with data obtained previously with RV [44], influenza A virus (IAV) infection of CFTR knock-down (KD) AECs showed robust pro-inflammatory response, as evidenced by the IL-8, IL-6, and IP10 gene transcripts" (line 151).

75) Please change "the dysregulation" to "dysregulation" (line 153).

76) Please replace "signaling" with "signalling" (lines 154, 157, 162, 180, 188, 199, 232, 260, 376, 386, 406 (2x), 408, 410, 479, 523, 524, 529, 607).

77) Please replace "cells (AECs)" with "AECs" (line 154).

78) Please change "airway epithelial cells" to "AECs" (lines 158, 475, 649).

79) It is not exactly clear what the authors mean by "less IFN-, IFN-–regulated protein CXCL10 (IP-10), a CXCR3 chemokine critically important in the development of a Th1 response to extracellular pathogens, and type III IFN-" in "Using different models of immortalised CF and non-CF pairs of AECs, the authors demonstrate that P. aeruginosa induced less IFN-, IFN-–regulated protein CXCL10 (IP-10), a CXCR3 chemokine critically important in the development of a Th1 response to extracellular pathogens, and type III IFN- in CF cells as compared with non-CF counterparts" (line 162)? Are they referring to induced protein expression? Please elaborate in the text.

80) Please replace "CXCL10 (IP-10)" with "CXCL10/IP-10" (line 164).

81) Please change "III" to "III interferon" (line 166).

82) Please replace "with" with "with their" (line 166).

83) Please change "As to PRRs, TLR1–10" to something like "The PRRs TLR1–10" (line 167).

84) Please replace "1 and 3" with "I and III" (line 168).

85) It is not entirely clear what the authors refer to as "airway Gram-negative microbes" in "Rather, it has been reported that TLR4, the lipopolysaccharide (LPS) PRR, is displayed at very low levels on the apical surface of CF AECs [52,53], possibly altering the response to airway Gram-negative microbes and contributing to chronic bacterial infection in CF airways" (line 171)? Please fix in the text.

86) Please change "and" to "and thereby" (line 174).

87) Please replace "AEC" with "AECs" (line 182).

88) Please change "hyper-activation" to "hyperactivation" (line 183).

89) Please replace "oxide (NO)" with "oxide" (line 190).

90) Please change "(NOS)-2" to "(NOS) 2" (line 190).

91) Please replace "defense" with "defence" (line 191).

92) Please change "bacteria, was" to "bacteria was" (line 191).

93) Please replace "AKT" with "Akt" (lines 192, 195, 328, 367).

94) Please replace "Fibrosis (NCF)" with "fibrosis" (line 198).

95) Please change "recapitulates" to something like "recapitulates immune" (line 199).

96) Please replace "NFKB1 (Nuclear Factor Kappa B Subunit 1)" with "nuclear factor kappa B subunit 1 (NFKB1)" (line 200).

97) Please change "nana, the Japanese word for 17)" to "nana (the Japanese word for 17)" (line 204).

98) Please replace "of" with "of the" (line 204).

99) Please provide reference for "IRF1,3,5, and 7 (involved in the innate immune 208 response phenotype and controlling expression of type-1 interferons upon viral infection)" mentioned in "Other pathways included caspase (CASP) 3 and CASP7 as effectors of apoptosis, CASP1, effector of pyroptosis (a highly pro-inflammatory cell death mechanism), regulators of actin cytoskeleton pathway (ACTN4, ARPC5, PFN, MYL12B, and 207 VCL), as well as IRF1,3,5, and 7 (involved in the innate immune 208 response phenotype and controlling expression of type-1 interferons upon viral infection)" (line 205).

100) Please change "CASP7 as" to "7 as the" (line 205).

101) Please replace "effector" with "the effector" (line 206).

102) Please change "pyroptosis (a highly pro-inflammatory cell death mechanism)" to "pyroptosis, a highly pro-inflammatory cell death mechanism" (line 206).

103) Please replace "7 (involved in the innate immune response phenotype, and controlling expression of type-1 interferons upon viral infection)" with "7, involved in the innate immune response phenotype, and controlling expression of type-1 interferons upon viral infection" (line 208).

104) Please change "phenotype and controlling" to "phenotype, controlling the" (line 209).

105) Please replace "AEC" with "airway epithelial cell" (line 212).

106) Please change "airway epithelial cell" to "AEC" (line 214).

107) Please replace "brushing, nasal brushing and" with "brushings, nasal brushings, and", "lines, IB3-1" with "lines IB3-1" in Table 1.

108) Please replace "Insulin Like Growth Factor" with "insulin-like growth factor" (line 218).

109) Please change "PI3K, phosphoinositide 3-kinase; RANK, Receptor Activator of Nuclear Factor Kappa B; SHIP1, Src homology 2 (SH2) domain containing inositol polyphosphate 5-phosphatase 1; Tollip, Toll-interacting protein; TOM1, target of Myb1; WISP1, WNT1-inducible-signaling pathway protein 1. ND: not determined" to "ND, not determined; PI3K, phosphoinositide 3-kinase; RANK, receptor activator of nuclear factor kappa B; SHIP1, Src homology 2 (SH2) domain containing inositol polyphosphate 5-phosphatase 1; Tollip, Toll-interacting protein; TOM1, target of Myb1; WISP1, WNT1-inducible signalling pathway protein 1." (line 220).

110) Please format "Src homology 2 (SH2) domain containing inositol polyphosphate 5-phosphatase 1" consistently with the rest of the text (line 220).

111) Please change "Insulin-Like Growth Factor" to "Insulin-like growth factor" (line 246).s

112) Please remove italics formatting from "Insulin-Like Growth Factor 1" (line 246).

113) Please remove italics formatting from "WNT1-inducible-signaling pathway protein 1" (line 246).

114) Please replace "lipopeptide, in" with "lipopeptide in" (line 264).

115) Please change "on" to "in" (line 265).

116) Please replace "lung as an attempt" with "lung" (line 266).

117) Please change "bronchoalveolar lavage fluid" to "bronchoalveolar lavage fluid (BALF)" (line 272) and "bronchoalveolar lavage fluid (BALF)" to "BALF" (line 502).

118) Please replace "line (CFBE41o−)" with "line CFBE41o−, a" (line 282).

119) Please change "IKKβ protein" to "IKKβ" (line 283).

120) Please replace "not be a guarantee of" with "warrant" (line 286).

121) Please provide reference for "Nevertheless, primary bronchial epithelial cells confirmed that AECs downregulate miR-199a-3p in CF" (line 287).

122) Please change "structure, without" to "structure without" (line 300).

123) Please replace "lipopolysaccharide" with "LPS" (line 315).

124) Please replace "PI3K/AKT" with "PI3K/Akt" (lines 324, 331, 334, 406, 524).

125) Please change "functions cell biology" to "functions" (line 325).

126) Please replace "polarization" with "polarization," (line 326).

127) Please change "signaling" to "signalling" (line 327).

128) Please replace "cascades, including" with "cascades including" (line 329).

129) Please provide reference for "Besides tissue-resident macrophages, neutrophils and monocytes are attracted into the lung under inflammatory conditions" (line 336).

130) Please change "as" to "as the" (lines 345, 533).

131) Please replace "(CCL2, also known as MCP-1)" with "(CCL2) (also known as MCP-1)" (line 345).

132) Please change "alveolar macrophages" to "AMs" (lines 366, 537).

133) Please replace "PI3K/Akt" with "the PI3K/Akt pathway" (line 368).

134) Would the authors please replace "properly" in "In CF macrophages, derived from peripheral blood monocytes, secretion of sCD14, IL-1β, IL-6, and TNF-α is heightened, and expression of CD11b and TLR-5 were profoundly decreased, highlighting that CF macrophages are unable to properly recognize pathogens and contribute to an unremittingly inflammatory state" using a more formal expression (line 371)?.

135) Please change "heightened and" to "heightened, while" (line 372).

136) Please replace "hyper-secretion" with "hypersecretion" (line 374).

137) Please change "causes" to something like "induces" (line 376).

138) Please replace "monocyte-derived macrophages" with something like "monocyte-derived macrophage counts" (line 382).

139) Please change "LPS" to "LPS exposure" (line 383).

140) Please replace "Mad-Related Protein" with "Mad-related protein" (line 385).

141) Please change "be due to" to something like "stem from" (line 395).

142) Please remove italics formatting from "TLR4" (line 405).

143) Please replace "have a" with something like "display" (line 408).

144) Please change "PI3K/pAKT" to "PI3K/pAkt" (line 408).

145) Please replace "TRAF6" with "TNF receptor-associated factor 6 (TRAF6)" (line 416) and "TNF Receptor Associated Factor 6" with "TFAF6" (line 642).

146) Please change "apoptosis and" to "apoptosis," (line 442).

147) Please replace "lung" with "lung," (line 445).

148) Please change "CF" to "cystic fibrosis" (line 452).

149) Please replace "plasma" with "the plasma" (line 454).

150) Please remove italics formatting from "PDLIM2PDZ and LIM domain 2" (line 463).

151) Please change "Suppressor Of Cytokine Signaling" to "suppressor of cytokine signalling (line 463).

152) Please remove italics formatting from "Suppressor Of Cytokine Signaling" (line 463).

153) Please replace "quantitative real-time PCR (qRT-PCR)" with "qRT-PCR", "bronchial epithelial cells (BECs)" with "BECs" in Table 2.

154) Please define abbreviation for "qRT-PCR", "BECs", "IL-18" in the footnote of Table 2.

155) Please change "the" to "while" (line 493).

156) Please replace "inflammation attenuation" with "the attenuation of inflammation" (line 494).

157) Please change "The" to "In the" (line 494).

158) Please replace "brushings as such" with "brushings" (line 495).

159) Please change "composition, comprised" to "composition comprised" (line 495).

160) Please replace "thereby indicating" with "indicating" (line 496).

161) Please change "Heme Oxygenase" to "heme oxygenase" (line 517).

162) Please replace "MiRNA-126-3p" with "miRNA-126-3p" (line 546).

163) Please change "INPP5D (Inositol Polyphosphate-5-Phosphatase D)/SHIP1" to "inositol polyphosphate-5-phosphatase D (INPP5D)/SHIP1" (line 548).

164) Please remove italics formatting from "Inositol Polyphosphate-5-Phosphatase D" (line 548).

165) Please replace "Fig." with "Figure" (line 557).

166) Please change "representing interaction" to "interaction" (line 562).

167) Please replace "MiRNAs" with "miRNAs" or "MicroRNAs" (line 589).

168) Please change "PI3K/Akt signaling, Wnt/β-catenin signaling, glucocorticoid receptor signaling" to "PI3K/Akt, Wnt/β-catenin, glucocorticoid receptor" (line 595).

169) Please replace "samples, with" with "samples with" (line 604).

170) Please change "miR-25-3p" to "miR-25-3p," (line 616).

171) Please replace "MiRNA" with "miRNA" (3x), "hsa-let-7c-5p" with "hsa-let-7c-5p,", "samples:" with "samples,", "miR-25-3p" with "miR-25-3p,", "exacerbation, with" with "exacerbation with" in Table 3.

172) Please define abbreviation for "HC" in the footnote of Table 3.

173) Please change "Distribution of CF-Associated miRNAs" to "distribution of cystic fibrosis-associated miRNAs" (line 628).

174) Please provide reference for "For miR-155-5p, chronic inflammatory characteristic of the CF airway microenvironment activates NF-κB signalling, driving its upregulation in epithelial cells and neutrophils; increased miR-155 suppresses SHIP1, thereby enhancing PI3K/Akt activation and contributing to the excessive production of pro-inflammatory mediators such as IL-8" (line 623).

175) Please provide reference for "miR-636 similarly shows elevated expression in bronchial epithelial cells and in circulating neutrophils from individuals with CF, likely reflecting a shared response to persistent inflammatory stress" mentioned in "miR-636 similarly shows elevated expression in bronchial epithelial cells and in circulating neutrophils from individuals with CF, likely reflecting a shared response to persistent inflammatory stress; this pattern suggests a potential role for miR-636 in coordinating inflammatory signalling across epithelial and innate immune compartments" (line 636).

176) Please replace "stress; this" with "stress. This" (line 638).

177) From "miR-146a was shared between monocytes/macrophages [125] and the EC miRNAs fraction" (line 639) is not unequivocally clear whether the authors mean to say "monocytes and macrophages" or "monocytes or macrophages"? Please correct.

178) Please provide reference for "miR-146a is a key regulator of innate immune signalling, particularly in monocytes and macrophages, where it negatively regulates NF-κB–dependent pro-inflammatory responses by targeting TNF Receptor Associated Factor 6 (TRAF6) and Interleukin 1 Receptor Associated Kinase 1 (IRAK1)" (line 640).

179) Please change "Interleukin 1 Receptor Associated Kinase" to "interleukin-1 receptor-associated kinase" (line 643).

180) Please provide reference for "Under chronic inflammatory conditions, miR-146a can be actively secreted into the extracellular space via exosomes or microvesicles, contributing to its detection in circulating EC-miRNAs and mediating intercellular modulation of inflammation" (line 643).

181) Please provide reference for "miR-223-3p is highly expressed in mature neutrophils" mentioned in "miR-223-3p was found in both AECs [141] and neutrophils [145], miR-223-3p is highly expressed in mature neutrophils, reflecting its role in myeloid differentiation and neutrophil function" (line 646).

182) Please replace "[145], miR-223-3p" with "[145]. miR-223-3p" (line 646).

183) Please provide reference for "Activated neutrophils can release miR-223 via microvesicles or exosomes, which can then be taken up by airway epithelial cells" (line 648).

184) Please provide reference for "Consequently, miR-223-3p becomes detectable in epithelial cells, indicating a neutrophil–epithelial intercellular miRNA transfer that may help regulate airway inflammation and epithelial responses" (line 649).

185) Please change "CF-associated" to "Cystic fibrosis-associated" (line 653).

186) From "A network of cells interacting each other via miRNAs and/or extracellular vesicles would it possible, however the appraisal of all the works we scrutinised in this Review, it appears that cell types considered are of single origin, either AECs or monocytes-macrophages or neutrophils" (line 658) is not unequivocally clear whether the authors mean to say "miRNAs and extracellular vesicles" or "miRNAs or extracellular vesicles"? Please fix.

187) "A network of cells interacting each other via miRNAs and/or extracellular vesicles would it possible, however the appraisal of all the works we scrutinised in this Review, it appears that cell types considered are of single origin, either AECs or monocytes-macrophages or neutrophils" (line 658) is not gramatically correct with respect to "interacting each other via miRNAs and/or extracellular vesicles would it possible, however the appraisal of all the works we scrutinised in this Review, it appears". Please rephrase.

188) Please replace "Review" with "review" (line 660).

189) "Nevertheless, it is tempting to speculate a miRNA signalling among these cell types involved in CF lung disease, a topic to be explored in vivo on freshly obtained samples from the airways or in vitro in co-culture systems" (line 661) is not grammatically correct with respect to "speculate a miRNA signaling". Please revise.

190) Please change "be" to something like "be best" (line 662).

191) Please replace "CF lung disease" with "CFLD" (line 668).

192) Please change "on how to gauge" to "into gauging" (line 674).

Reviewer 3 Report

Comments and Suggestions for Authors

The reviewer thaks the authors for providing additional information on miR in CF in the revised version, engaging in network analysis and for thoroughly editing the text. This will now be a highly informative, albeit specialized, review and a good contribution to the field.

Round 3

Reviewer 1 Report

Comments and Suggestions for Authors

Major point:

There seems to be way too many small issues hampering clear understanding of the discussed topics. The authors are therefore advised to thoroughly revisit their manuscript (see the minor point list below).

Minor points:

1) Please change "Namra Sajid1,2§, Piera Soccio3, Pasquale Tondo3,4, Donato Lacedonia3,4, Sante Di Gioia1" to "Namra Sajid 1,2§, Piera Soccio 3, Pasquale Tondo 3,4, Donato Lacedonia 3,4, Sante Di Gioia 1" (line 4).

2) Please replace "inflammatory and immune responses becoming dysregulated and determining more harm than benefit" with something like "dysregulated inflammatory and immune responses" (line 16).

3) Please change "molecular components (cytokines, chemokines, signal transduction pathways, transcription factors)" to "molecular (cytokines, chemokines, signal transduction pathways, transcription factors) components" (line 19).

4) Please replace "CF lung disease" with "CFLD" (lines 24, 39, 91, 110, 111, 919, 1089, 1346).

5) Please change "CF lung disease, including" to "CFLD including" (line 26).

6) Please replace "presentation, and" with "presentation and" (line 27).

7) Please provide reference for "Dysfunction of or the lack of CF transmembrane conductance regulator (CFTR), a chloride and bicarbonate channel, leads to abnormal hydration and viscoelasticity of airway mucus" (line 40).

8) Please change "Dysfunction of" to "Dysfunction" (line 40).

9) Please replace "CF transmembrane conductance regulator (CFTR), a chloride and bicarbonate channel, leads" with "the CF transmembrane conductance regulator (CFTR) chloride and bicarbonate channel leads" (line 40).

10) Please change "and fueling" to "fuelling" (line 43).

11) Please replace "surface, not allowing a" with something like "without" (line 88).

12) "Altered repair of airway epithelium contributes to lung remodelling ensuing destruction of proximal airways and ultimately lung function insufficiency" (line 88) does not seem to be grammatically correct with respect to "lung remodelling ensuing destruction". Please revise.

13) Please change "of" to "of the" (lines 88, 220, 602, 708, 711, 1095, 1231).

14) Please replace "unsolvable inflammatory response is the" with "inflammatory response is the unsolvable" (line 91).

15) Please change "to" to "to the" (lines 92, 593).

16) Please replace "stimulates" with "stimulate" (line 93).

17) Please change "interleukin 8" to "interleukin 8 (IL-8)" (line 94).

18) Please replace "in" with "in the" (line 95).

19) "increase" could be changed to something like "potentiate" (line 95).

20) Please replace "level, specifically" with "level by specifically" (line 98).

21) Please change "MiRNAs" to "miRNAs" (line 99).

22) Please provide reference for "For this reason, miRNAs play a critical role in many diseases characterised by dysregulation of miRNA expression" (line 100).

23) Please change "processes, as" to "processes as" (line 104).

24) Please replace "CF" with "CFLD" (line 105).

25) Please change "are considered to" to something like "may" (line 110).

26) Please replace "of cell" with "of the cell" (line 111).

27) Please change "cellin" to something like "cell population in" (line 112).

28) Please replace "https://pubmed.ncbi.nlm.nih.gov/" with "https://pubmed.ncbi.nlm.nih.gov" (line 118).

29) Please change "https://www.scopus.com/" to "https://www.scopus.com" (line 118).

30) Please replace "such" with "respective" (line 136).

31) It is not exactly clear what the authors mean by "control infections prominently by respiratory viruses" in "The essential feature of type 1 AEC responses is the induction of type I and III interferons (IFNs) and interferon-stimulated genes (ISGs) to control infections prominently by respiratory viruses (line 136)? Are they referring to the predominant control of respiratory infections?

32) Please change "mucin (MUC) 5AC" to "mucin 5AC (MUC5AC)" (line 177).

33) Please replace "though" with "through" (line 178).

34) Please change "Domain Containing" to "domain-containing" (line 180).

35) Please replace "interleukin (IL)-13" with "IL-13" (line 184).

36) Please change "AEC type 3" to "Type 3 AEC" (line 184).

37) Please replace "(NLRs), and" with "(NLRs) and" (line 187).

38) "these responses"/this response" appears twice in "Cytokines involved in the amplification of these responses are IL-22 and multiple IL-17 family members, which operate to recruit neutrophils, the effector cell type in this type of immunity [40,41]. Besides neutrophil recruitment, induction of inflammatory cytokines such as IL-1, IL-6, and the tumor necrosis factor α (TNF-α) and the induction of antimicrobial peptides (AMPs) is the other pillars of this response" (line 189). Please fix.

39) From "Cytokines involved in the amplification of these responses are IL-22 and multiple IL-17 family members, which operate to recruit neutrophils, the effector cell type in this type of immunity" (line 189) is not explicitly clear what the authors refer to as "this type of immunity"? Please fix.

40) Please change "tumor" to "tumour" (line 192).

41) Please replace "and" with "as well as" (line 193).

42) Please change "is" to "are" (line 193).

43) Please replace "AEC dysfunctional state" with "The dysfunctional state of AECs" (line 198).

44) Please change "(in the absence of infection)" to "in the absence of infection" (line 198).

45) Please replace "secretion" with "secretion," (line 200).

46) Please change "airway surface liquid (ASL) pH" to "the pH of airway surface liquid (ASL)" (line 201).

47) Please replace "AECs type 1" with "type 1 AECs" (line 203).

48) Please provide reference for "RV serotypes are classified as major or minor group depending on the surface receptor used to infect target cells" (line 205).

49) Please provide reference for "Infection (1 hr) with major group RV16 of primary CF AECs resulted in a trend towards a diminished IFN response at the level of IFN-λ1, IFN-λ2/3 and IFN-β, the PRRs TLR3, retinoic acid-inducible gene I (RIG-I), as well as melanoma differentiation-associated protein 5 (MDA5) and ISGs in comparison to healthy AECs" (line 206).

50) Please change "IFN-λ2/3" to "IFN-λ2/3," (line 208).

51) Please replace "stimulation by polyinosinic:polycytidylic acid (poly(I:C)) – a strong IFN inducer – determined" with something like "the potent IFN inducer polyinosinic:polycytidylic acid (poly(I:C)) stimulated" (line 212).

52) Please define abbreviation for "HBEs" (line 217).

53) Please change "knock-down" to "knockdown" (line 222).

54) Please replace "DNA" with something like "DNA sequences" (line 265).

55) The authors mention "the authors" in "Using different models of immortalised CF and non-CF pairs of AECs, the authors demonstrate, at the mRNA and protein levels, that P. aeruginosa induced less IFN-, IFN-–regulated protein CXCL10/IP-10, and the type III interferon IFN- in CF cells as compared with their non-CF counterparts" (line 270) despite no prior mention of any specific author. Please fix.

56) Please change "demonstrate, at the mRNA and protein levels, that" to "demonstrate at the mRNA and protein levels that" (line 271).

57) Please replace "induced" with "induces" (line 272).

58) Please change "IFN-" to "IFN- expression" (line 273).

59) Please replace "cells, and it" with "cells. It" (line 278).

60) Please change "TLR4, the lipopolysaccharide (LPS) PRR, is" to "the lipopolysaccharide (LPS) PRR TLR4 is" (line 280).

61) From "Intracellular signal pathways are deregulated in AECs and other immune cells such as monocytes and macrophages, and CFTR is likely to play an important role both for its role of chloride channel and regulator of various other ion channels, transporters, and receptors" (line 285) is not explicitly clear the role that "CFTR is likely to play"? Role in what?

62) The authors mention "this hyperactivation" in "The NF-B pathway is basally dysregulated in CF primary AECs obtained from nasal polyps and this hyperactivation correlated with increased IL-8 secretion" (line 289) despite no prior mention of any activation/hyperactivation. Please fix.

63) Please replace "are" with "were" (line 292).

64) Please change "is" to "was" (lines 292, 601).

65) Please replace "mitogen-activated protein kinase/extracellular signal-regulated kinase (MAPK/ERK)" with "mitogen-activated protein kinase (MAPK)/extracellular signal-regulated kinase (ERK)" (line 293).

66) Please change "in the" to "in" (line 296).

67) Please replace "did not" with "failed to" (line 299).

68) Please change "Fibrosis" to "fibrosis" (line 305).

69) Please replace "Factor Kappa B Subunit" with "factor kappa B subunit" (line 307).

70) Please change "ju-nana (the Japanese word for 17) (JUN),FOS and JUN being part of the AP-1 complex).Other" to "jū-nana (the Japanese word for 17) (JUN),FOS and JUN being part of the AP-1 complex). Other" (line 310).

71) Please replace "of actin" with "of the actin" (line 313).

72) Please change "7, involved" to "7 involved" (line 491).

73) Please replace "type-1" with "type 1" (line 491).

74) Please change "brushing" to "brushings" (2x) in Table 1.

75) Please define abbreviation for "IL-8" in the footnote to Table 1.

76) Please replace "Abbreviations: AECs" with "AECs" (lines 510, 1077).

77) Please change "air liquid" to "air-liquid" (lines 510, 1077).

78) Please replace "domain containing" with "domain-containing" (lines 513, 522).

79) Please replace "cell, IB3-1/S9" with "cells IB3-1/S9" (line 520).

80) Please change "(SHIP1), an inositol phosphate phosphatase, in" to "(SHIP1) in" (line 523).

81) Please replace "hyper-expression" with "hyperexpression" (line 525).

82) The authors mention "the authors" in "In a subsequent study in CF primary bronchial cells differentiated in airliquid interface (ALI) cultures, the authors found that miR-636 overexpression by mimic miRNA repressed interleukin-1 receptor 1 (IL1-R1) and inhibitor of nuclear factor kappa B kinase subunit beta (IKK- and increased TNF receptor sub-family 11A (TNFRSF11A, also called Receptor Activator of Nuclear Factor Kappa B (RANK), protein expression in CF cells, resulting in overall decrease activation of the NF-κB pathway" (line 528) despite no prior mention of any specific author. Please fix.

83) Please change "airliquid interface (ALI)" to "ALI" (line 528).

84) Please replace "repressed" with "repressed the" (line 530).

85) Please change "(IKK-)" to "(IKK-)," (line 531).

86) Please replace "(TNFRSF11A, also called Receptor Activator of Nuclear Factor Kappa B (RANK)" with "(TNFRSF11A) (also called receptor activator of nuclear factor kappa B (RANK))" (line 531).

87) Please change "In a deep sequencing study of miRNome, it was" to "A deep sequencing study of the miRNome" (line 537).

88) Please replace "brushings and" with "brushings, and" (line 538).

89) Please change "WNT1-inducible-signalling" to "WNT1-inducible signalling" (line 539).

90) Please replace "the growth factor IGF1" with "IGF1" (line 542).

91) Would the authors please replace "proper" in "On the other hand, miR-101-3p and its isoforms were also deregulated in CF cells. miR-101-3p and miR-181a-5p silencing stimulated increased WISP1 levels and proper wound healing" using a more formal expression (line 543)?

92) Please change "MiR-126" to "miR-126" (line 549).

93) Please replace "are" with "were the" (line 550).

94) Please change "and" to "and the" (lines 550, 1018, 1112).

95) Please replace "over-expression" with "overexpression" (line 554).

96) Please change "protein" to "protein levels" (line 554).

97) Please replace "inflammation, with TOM1 increased expression" with something like "inflammation accompanied by the increased expression of TOM1" (line 557).

98) Please change "burden in this condition" to "burden" (line 558).

99) Please replace "MiR-17" with "miR-17" (line 560).

100) Please change "BECs." to "BECs" (line 583).

101) Please replace "be possible" with something like "occur" (line 587).

102) Please change "suggest" to "suggested" (line 588).

103) Please replace "in in" with "in" (line 590).

104) Please change "to the" to "to" (line 594).

105) Please replace "a CF" with "the CF" (line 594).

106) Please change "suppressed" to "suppressed the" (line 595).

107) Please replace "thus the" with "thus" (line 598).

108) Please change "MiR-93" to "miR-93" (line 601).

109) Please replace "(AMs), and" with "(AMs) and" (line 608).

110) Please change "of the" to "of" (line 609).

111) Please replace "not damage alveoli and disrupt" with something like "prevent damage to alveoli and disruption of the" (line 612).

112) Please change "(TNF-α, IL-1, and IFN-γ)" to "TNF-α, IL-1, and IFN-γ" (line 615).

113) Please replace "normal" with "the" (line 620).

114) Please change "mechanisms are" to "mechanism is" (line 621).

115) Please replace "in the production of" with "to produce" (line 622).

116) Please change "‘classically activated macrophages’" to ""classically activated macrophages"" (line 625).

117) Please replace "‘alternatively activated macrophages’" with ""alternatively activated macrophages"" (line 625).

118) Please change "response, such" to "response such" (line 626).

119) Please replace "M2 macrophages subsets exert" with something like "A subset of M2 macrophages exerts" (line 630).

120) Please change "dendritic cells" to "DCs" (line 683).

121) Please replace "chemokines, including" with "chemokines including" (line 684).

122) Please change "expands, and" to "expands and" (line 692).

123) Please replace "CD11b, where CD stays for cluster of differentiation)" with "cluster of differentiation 11b (CD11b)" (line 704).

124) Please narrow the span of the text between lines 710 and 1025 so that its paragraph width becomes consistent with the rest of the surrounding text.

125) Please replace "peripheral blood monocytes" with "PBMCs" (line 710).

126) Please change "CD14(sCD14)" to "CD14 (sCD14)" (line 711).

127) Please replace "signalling upregulation" with "signalling" (line 716).

128) Please change "recruitment" to something like "mobilisation" (line 905).

129) Please replace "And" with "and" (line 906).

130) Please change "peripheral blood monocyte-derived" to "PBMC-derived" (line 911).

131) Please replace "with" with "with the" (line 914).

132) Please change "macrophages, could" to "macrophages could" (line 916).

133) Please replace "cell autonomous immune dysfunctions" with something like "types of cell autonomous immune dysfunction" (line 918).

134) Please change "disease, which make" to something like "disease and render" (line 919).

135) Please replace "MiR-199a-5p" with "miR-199a-5p" (line 923).

136) "blunted" appears twice in "This axis is regulated by PI3K/Akt signalling, which is blunted in CF macrophages, leading to increased level of miR-199a-5p and decreased levels of CAV1 [129]. In summary, CF macrophages display blunted PI3K/pAkt signalling in response to TLR4 activation, which leads to accumulation of miR-199a-5p, decreased CAV1 expression, ultimately leading to ineffective negative feedback of TLR4 signalling and hyperinflammation" (line 926). Please correct.

137) Please change "level" with "levels" (lines 928, 1095).

138) Please replace "peripheral blood monocytes-derived" with "PBMC-derived" (line 932).

139) Please change "inflammation, and was further studied" to "inflammation" (line 936).

140) Please replace "MiR-146a" with "miR-146a" (line 936).

141) Please change "TNF" to "the TNF" (line 937).

142) Please replace "reduces" with "reduced" (line 938).

143) Please change "on" to "in" (line 941).

144) Please replace "non-CF one" with "non-CF" (line 944).

145) Please change "as in" to "as" (line 955).

146) Please replace "neutrophils extracellular traps" with "neutrophil extracellular traps (NETs)" (line 989) and "neutrophil extracellular traps (NETs)" with "NETs" (line 993).

147) Please change "via the" to "via" (line 993).

148) Please replace "this last" with "this" (line 996).

149) Please change "MiR-636" to "miR-636" (line 1006).

150) Please replace "proteins, as" with "proteins as" (line 1008).

151) Please change "males, with" to "males with" (line 1015).

152) Please replace "cells and were" with "cells" (line 1023).

153) Please change "using as" to "using" (line 1023).

154) Please replace "Cell line/Cell culture" with "Cell line/cell culture", "utilised as such for" with "utilised for", "expression." with "expression", "cells." with "cells", "compared to control" with "compared to the control", "deregulated miRNAs" with "deregulated miRNA", "the results of" with "the results demonstrating", "and members of" with "and the members of the", "Isoform of miRNA such as miR-101-3p altered form" with "miR-101-3p", "WISP1 target" with "WISP1,", "proliferation/migration" with "proliferation, migration,", "process." with "process", "The results showed that several" with "Several", "mRNA/protein level" with "mRNA and protein levels", "increased in ALI" with "increased in the ALI", "CF." with "CF", "dysregulated biological process" with "dysregulated biological processes", "inflammation." with "inflammation", "The airway" with "Airway", "that likely" with "that are likely", "target i.e." with "target, i.e.", "study identifies" with "study identifies the", "target." with "target", "The upregulated" with "Upregulated", "by binding with" with "by binding to a", "the CFTR" with "CFTR" in Table 2.

155) Please change "AECs:" to "AECs," (line 1077).

156) Please replace "IL6R:" with "IL6R," (line 1078).

157) Please change "IL-11:" to "IL-11," (line 1078).

158) Please replace "NCF:" with "NCF," (line 1079).

159) Please change "TOM1:" to "TOM1," (line 1079).

160) Please change "a target" to "as a target the" (line 1081).

161) Please replace "(the subunit gp130" with something like "which represents the gp130 subunit" (line 1081).

162) Please change "hyper-responsive" to "hyperresponsive" (line 1083).

163) Please replace "IL-6 classic" with "the classic IL-6" (line 1083).

164) Please change "will also operate" to "also operates" (line 1084).

165) It is not exactly clear what the authors refer to as by "increase of TOM1mRNA" in "Oglesby et al. [75] already demonstrated that CF cells show an increase of TOM1mRNA , a negative regulator of the NF-B pathway, in consequence of miR-126 
downregulation" (line 1086)? Are they referring to expression levels? Please fix.

166) It is not clear what the authors mean by "the positive regulation of IL6ST" in "This effect on TOM1 expression would possibly exert a compensation for the hyperinflammation occurring in CF lung disease, while the positive regulation of IL6ST is in line with the attenuation of inflammation" (line 1086)? Moreover, the link between "TOM1 expression" and "the positive regulation of IL6ST" is rather elusive. Please correct.

167) Please replace "[75] already" with "[75]" (line 1086).

168) Please change "TOM1mRNA" to "TOM1 mRNA" (line 1087).

169) Please replace "in" with "as a" (line 1087).

170) Please change "hyperinflammation" to something like "hyperinflammation phenotype" (line 1089).

171) Please replace "the work by Oglesby et al. [75], freshly isolated bronchial brushings were used, likely containing a mixed cell type composition comprised of immune cells, indicating that miR-126 downregulation could be either in immune cells or that immune cells could exert a regulatory effect on AECs" with something like "this work, freshly isolated bronchial brushings were used, likely containing a mixed cell type composition comprised of immune cells, indicating that miR-126 downregulation could occur either in immune cells or that immune cells could exert a regulatory effect on AECs [75]" (line 1090).

172) Please change "cause not only suppression of CFTR mRNA and protein level" to something like "not only suppress the mRNA and protein levels of CFTR" (line 1095).

173) Please replace "has" with "has as" (line 1096).

174) Please change "witnessing" to something like "attesting to" (line 1097).

175) Please change "extracellular vesicles" to "extracellular vesicles (EVs)" (line 1101) and "Extracellular vesicles (EVs)" with "EVs" (line 1346).

176) Please replace "with the many" with something like "within the plethora of" (line 1101).

177) Please change "both CF cell lines, CFBE41o- and IB3" to "the CFBE41o- and IB3 cell lines" (line 1105).

178) Please provide reference for "BTB domain and CNC homolog 1 (BACH1) (), a target of miR-155-5p, is a transcription factor and a master regulator of antioxidant systems" (line 1106).

179) Please replace "(BACH1) ()" with "(BACH1)" (line 1106).

180) Please change "in" to "in the" (line 1108).

181) Please replace "(HO-1) ()" with "(HO-1)" (line 1113).

182) Please change "less" to something like "reduced" (line 1113).

183) Please replace "MiR-155-5p" with "miR-155-5p" (line 1116).

184) Please change "impaired" to something like "impaired the activity of" (line 1116).

185) Please replace "Src homology 2 (SH2) domain containing inositol polyphosphate 5-phosphatase 1 (SHIP1) (1)" with "SHIP1" (line 1116).

186) Please change "air-liquid interface" to "ALI" (line 1234).

187) Please replace "up regulation" with "upregulation" (line 1236).

188) Please change "WNT1 inducible signalling pathway (WISP1)" to "WISP1" (line 1237).

189) Please remove italics formatting from "signalling" (line 1238).

190) Please replace "key regulator" with "the key regulators" (line 1240).

191) Please change "the insulin-like growth factor 1 (IGF-1)" to "IGF1" (line 1240).

192) It is not exactly clear what the authors mean by "low IGF-1 protein" in "This finding suggests that compromised innate immunity in CF lungs
is not solely due to epithelial defects but also involves immune cell dysfunction driven by low IGF-1 protein" (line 1245)? Are they referring to the expression level? Please revise.

193) Please replace "the 6" with "the" (line 1248).

194) Please change "by" to "by the" (line 1250).

195) Please replace "build" with "built" (line 1250).

196) Please change "miRNAs, and" to "miRNAs and" (line 1253).

197) Please replace "MiR-126-3p" with "miR-126-3p" (line 1253).

198) Please change "targeting" to "targets" (line 1253).

199) Please replace "TOM1, while" with "TOM1 while" (line 1254).

200) Please change "Inositol Polyphosphate-5-Phosphatase" to "the inositol polyphosphate-5-phosphatase" (line 1255).

201) Please format "INPP5D" using italics (line 1256).

202) Please format "SHIP1" using italics (line 1256).

203) Please replace "are not showing" with something like "do not display" (line 1257).

204) Please change "Three miRNAs, miR-145-5p" to "miR-145-5p" (line 1259).

205) Please replace "miR-101-3p, directly" with "miR-101-3p directly" (line 1259).

206) Please change "target and" to "target, and" (line 1281).

207) "this scheme, miR-145-5p is not anymore present since it did have any interaction with target genes accordingly with miRTargetLink (Figure 1)" (line 1292) does not seem to be grammatically correct with respect to "accordingly with miRTargetLink". Please rephrase.

208) Please replace "did" with "did not" (line 1293).

209) Please change "and then" to "hence" (line 1296).

210) Please replace "[74], and" with "[74] and" (line 1297).

211) Please change "shown" to something like "depicted" (line 1298).

212) Please replace "PI3K/Ak" with "PI3K/Akt", "hyper-inflammation" with "hyperinflammation" in Figure 2.

213) Please change "cells. Abbreviations of genes in the text" to "cells" (line 1307).

214) Please replace "Ideozu et al. [158] identified by microarray profiling 11" with "Using microarray profiling, Ideozu et al. [158] identified eleven" (line 1310).

215) Please change "miRNAs" to "miRNAs that" (line 1311).

216) Please replace "transduction, including pathways such as" with "transduction including the" (line 1314).

217) Please change "signalling" to "signalling pathway" (line 1315).

218) Please narrow the span of the text between lines 1319 and 1345 so that its paragraph width becomes consistent with the rest of the surrounding text.

219) From "Personalised medicine is the next big frontier in human disease and is also pursued in CF as identification of those groups with a defined genotype which should be treated with specific therapies" (line 1319) is not clear what the authors mean by "those groups with a defined genotype"? Please corroborate in the text.

220) Please replace "EV in sputum" with "EVs in sputum", "6-8" with "6–8", "CF." with "CF", "microRNAs" with "miRNAs", "miR-193a-5p." withv"miR-193a-5p" in Table 3.

221) Please change "Abbreviations: EBC:" to "EBC," (line 1360).

222) Please replace "EV:" with "EV," (line 1360).

223) Please change "FEV1:" to "FEV1," (line 1360).

224) Please replace "FVC:" with "FVC," (line 1360).

225) Please change "PE:" to "PE," (line 1361).

226) Please replace "fibrosis-Associated" with "fibrosis-associated" (line 1363).

227) Please change "CF-typical" to "CF-typical," (line 1366).

228) Please replace "chronic" with "the chronic" (line 1367).

229) Please change "MiR-223-3p" to "miR-223-3p" (line 1382).

230) Please replace "Monocytes/Macrophages" with "Monocytes/macrophages" in Table 4.

231) Please define abbreviation for "EC" in Table 4.

232) Please replace "Review" with "review" (line 1395).

233) Please change "extracellular vesicles-based" to "EV-based" (line 1396).

234) Please replace "macrophages" with "macrophages," (line 1397).

235) Please change "monocytes–macrophages" to "monocytes, macrophages" (lines 1402, 1412).

236) Please replace "MiRNA" with "miRNA" (line 1410).

237) Please change "cells (AECs)" to "cells" (line 1413).

Round 4

Reviewer 1 Report

Comments and Suggestions for Authors

Major points:

1) "F508del-CFTR homozygous CFTE29o− tracheal, CFBE41o− and IB3 BECs" mentioned in "MiR-17 was significantly decreased in vivo in CF bronchial brushings while overexpression of miR-17 in CF AECs (F508del-CFTR homozygous CFTE29o− tracheal, CFBE41o− and IB3 BECs) reduced the level of IL-8 protein under basal conditions and upon stimulation with LPS, Pseudomonas-conditioned medium, or bronchoalveolar lavage fluid (BALF) obtained from CF patients" (line 295) is way too similar to the expression "F508del-CFTR homozygous CFTE29o− tracheal, CFBE41o− and/or IB3 bronchial epithelial cells" found in "Overexpression of miR-17 inhibited basal and agonist-induced IL-8 protein production in F508del-CFTR homozygous CFTE29o− tracheal, CFBE41o− and/or IB3 bronchial epithelial cells" in reference [80] (Oglesby, I.K.; Vencken, S.F.; Agrawal, R.; Gaughan, K.; Molloy, K.; Higgins, G.; McNally, P.; McElvaney, N.G.; Mall, M.A.; 923 Greene, C.M. miR-17 overexpression in cystic fibrosis airway epithelial cells decreases interleukin-8 production. Eur Respir 924 J 2015, 46, 1350-1360). Please rephrase.

2) The data availability statement "All data presented in Table 2 are publicly available on GEO DataSets" (line 755) does not seem to refer to the author original dataset.

Minor points:

1) Please change "macrophages" to "macrophages," (line 25).

2) Please replace "presentation" with "presentation," (line 31).

3) Please change "of" to "of the" (lines 45, 315, 397, 563).

4) Please replace "with" with "to" (line 68).

5) "target" could be changed to "cognate" (line 68).

6) Please replace "MiRNAs" with "miRNAs" (line 70).

7) Please change "regulation of" to "regulating" (line 74).

8) Please provide reference for "Altered expression of specific miRNAs have been observed in CF" mentioned in "Altered expression of specific miRNAs have been observed in CF, potentially creating link between CFTR mutations and the pro-inflammatory phenotype" (line 76).

9) Please change "creating" to "creating a" (line 77).

10) Pease provide reference for "Despite the relevance of neutrophils in causing damage to the CF airways, other innate immunity cell types such as monocytes and macrophages and non-immune cells such as 
epithelial cells (line 79).

11) Please replace "causing damage" with something like "damaging" (line 79).

12) Please change "in" to "in the" (line 110).

13) Please replace "sensing chitin" with "chitin sensing" (line 120).

14) Please change "PRR" to "PRR protein" (lines 121, 187, 192).

15) Please replace "recognize" with "recognise" (lines 123, 403).

16) Please change "Type" to "The type" (line 126).

17) Please replace "oligomerization" with "oligomerisation" (line 128).

18) Please replace "(NLRs)" with "(NLRs)," (line 129).

19) Please change "operate t" to something like "recruit the effector neutrophils" (line 132).

20) Please provide reference for "Besides neutrophil recruitment, induction of inflammatory cytokines such as IL-1, IL-6, and the tumour necrosis factor α (TNF-α) as well as the induction of antimicrobial peptides (AMPs) are the other pillars of the type 3 immunity" (line 133).

21) Is "As a primary consequence, CFTR dysfunction results in defective bicarbonate secretion, which in turn reduces the pH of airway surface liquid (ASL), impairing the activity of AMPs and mucus properties" (line 142) is not semantically correct as it is difficult to imagine that "mucus properties" can be impaired. Do the authors mean to say "impairing the activity of AMPs and mucus secretion" instead?

22) Please replace "AEC" with "AECs" (line 147).

23) Please change "with" to "with the" (line 151).

24) Please replace "the IFN" with "IFN" (line 155).

25) Please change "upon" to "upon the" (line 155).

26) Please replace "of interferon" with "of the interferon" (line 157).

27) Please change "though" to "through" (line 179).

28) Please replace "in CF cells has been reported" with "has been reported in CF cells" (line 181).

29) Please change "it was demonstrated at the mRNA and protein levels that P. aeruginosa induces less IFN-, IFN-–regulated protein CXCL10/IP-10, and the type III interferon IFN-" to something like "P. aeruginosa was demonstrated to induce less IFN-, IFN-–regulated protein CXCL10/IP-10, and the type III interferon IFN- mRNA and protein" (line 183).

30) Please replace "and" with "and the" (line 187).

31) Please change "participating" to "to participate" (line 188).

32) Please provide reference for "However, some studies deny a higher expression of these immune receptors in CF vs. non-CF cells" (line 189).

33) Please replace "macrophages, and CFTR" with something like "macrophages. To this end, CFTR" (line 199).

34) Please change "play" to "have" (line 199).

35) Please replace "as" with "as a" (line 200 2x).

36) Please change "correlated" to "correlates" (line 205).

37) Please replace "when primary CF AECs were exposed to P. aeruginosa, there was activation of NF-κB, which drives the expression of IL-8" with something like "exposure of primary CF AECs to P. aeruginosa let to the activation of the NF-κB pathway and downstream expression of IL-8" (line 205).

38) Please change "found to be involved" to something like "implicated" (line 209).

39) Please replace "identified, i.e." with "identified as" (line 214).

40) Please change "localization [72]" to "localisation" (line 218).

41) Please replace "to create" with something like "the generation of" (line 222).

42) Please change "RELB (all part of the NF-B complex)" to "RELB, all part of the NF-B complex" or "RELB, part of the NF-B complex" (line 225).

43) Please replace "(FOS)" with "(FOS)," (line 225).

44) Please change "FOS" to "with FOS" (line 227).

45) Please replace "controlling" with "controlling the" (line 232).

46) Please define abbreviation for "Akt, IL-6, NF-kB" in the footnote to Table 1.

47) Please change "causing overexpression of" to "overexpressing" (line 249).

48) Please replace "pathway" with "pathways" (line 255).

49) Please change "stabilization" to "stabilisation" (lines 256, 316).

50) Please replace "by" with "by a" (line 260).

51) Please change "miR-636" to "the miR-636" (line 265).

52) Please replace "A" with something like "In addition, a" (line 266).

53) From "miR-101-3p and miR-181a-5p silencing stimulated increased WISP1 levels and wound healing" (line 276) is not explicitly clear whether the authors refer to mRNA or protein WISP1 levels? Please elaborate in the text.

54) Please change "Targeted by miR-126 were the target of" to "miR-126 targeted" (line 283).

55) Please replace "(Tollip))" with "(Tollip)" (line 284).

56) Please change "proinflammatory" to "pro-inflammatory" (lines 293, 503).

57) Please replace "MiR-17" with "miR-17" (line 294).

58) Please change "tracheal" to "tracheal cells" (line 296).

59) Please replace "CFBE41o−" with "CFBE41o−," (line 296).

60) Please change "IB3" to "IB3-1" (lines 296, 549).

61) Please replace "immortalized" with "immortalised" (line 301).

62) Please provide reference for "CF lung tissues are a mixed composition of non-immune cells (i.e. AECs) and immune cells as well as other cell types (i.e. fibroblasts and endothelial cells)" mentioned in "CF lung tissues are a mixed composition of non-immune cells (i.e. AECs) and immune cells as well as other cell types (i.e. fibroblasts and endothelial cells), thus regulation of miR-199a-3p in lung explants would not warrant specific AEC expression" (line 310).

63) "Nevertheless, primary BECs confirmed that AECs downregulate miR-199a-3p in CF" (line 312) is not 100% correct with respect to "BECs confirmed that AECs" as the only reason to claim this would be to inform the readers that "BECs" are "AECs". Please fix.

64) Please change "primary" to "experiments in primary" (line 313).

65) Please replace "MiR-93" with "miR-93" (line 314).

66) Please change "interstitial" to "interstitial (IMs)" (line 321) and "Interstitial macrophages (IMs)" to "IMs" (line 332).

67) Please define abbreviation for "PBMCs" (line 400).

68) Please replace "were" with "is" (line 402).

69) Please change "macrophages" to "macrophage" (line 412).

70) Please replace "imbalance, together" with "imbalance together" (line 424).

71) Please change "CF, was" to "CF was" (line 425).

72) Please replace "MiR-199a-5p" with "miR-199a-5p" (line 434).

73) Please change "MiR-146a" to "miR-146a" (line 449).

74) Please replace "Of" with "Of the" (line 457).

75) Please change "non-CF" to something like "non-CF subjects" (line 459).

76) Please replace "TGF-, is" with "TGF- is" (line 462).

77) Please change "neutrophils" to "neutrophil" (line 471).

78) Please replace "upon to" with something like "in" (line 475).

79) Please change "upregulate" to "upregulate the" (line 485).

80) Please replace "and" with "as well as" (line 487).

81) Please change "MiR-636" to "miR-636" (lines 490, 675).

82) Please replace "functions, i.e." with "functions such as" (line 491).

83) Please change "the blood neutrophils of CF" to "CF blood neutrophils" (line 494).

84) Please replace "potent" with "the" (line 496).

85) Please change "the two" to "the" (line 500).

86) Please replace "accomplish" with "obtain" (line 506).

87) From "From 67 studies found, three which were complying with AECs were extracted" (line 509) is not entirely clear what the authors mean by "complying with AECs"? Please fix.

88) Please change "three" to "3" (line 510).

89) Please replace "airway epithelial cells" with "AECs" (line 512).

90) Please change "IB3" to "IB3-1", "AECs, CFBE41o-; 1B3 Immortalised non-CF AECs, 16HBE14o-" to "AECs CFBE41o- and 1B3-1, immortalised non-CF AECs 16HBE14o-", "(CF vs NCF);" with "(CF vs NCF)," (3x), "Immortalised CF AECs, CFBE41o- Immortalised non-CF AECs, BEAS 2B; 16 HBE14o-" to "immortalised CF AECs CFBE41o-, immortalised non-CF AECs BEAS-2B and 16HBE14o-", "acted as" to "acting as", "the target gene" to "its target gene", "The results showed that five" to "Five", "miR-577-5p" to "and miR-577-5p", "down-regulated" to "downregulated", "MiR-101-3p" to "miR-101-3p", "WISP1" with "WISP1,", "migration" to "migration,", "healing process" to "healing", "e.g." to "e.g.,", "miR-494, miR-509-3p" to "miR-494, and miR-509-3p", "overexpressed in the" to "overexpressed in", "CFTR mRNA and protein levels" to "CFTR mRNA and protein levels.", "expression, that" to "expression that", "the target, i.e." to "their target" in Table 2.

91) "511 target genes of dysregulated miRNA evidenced IL-11, IL6R, and IL-18 genes" is not semantically correct in "511 target genes of dysregulated miRNA evidenced IL-11, IL6R, and IL-18 genes, which are related to lung and airway epithelium" in Table 2 as it is difficult to imagine that "target genes" can evidence "genes". Please revise.

92) It is not clear what the authors refer to as the "mMiR-101-3p altered form" in "mMiR-101-3p altered form along with miR-181a-5p directly impact the regulation of WISP1 which causes cell proliferation, migration and wound healing process" in Table 2? Please corroborate in the text.

93) Please specify the "specific target" mentioned in "Upregulated expression of different miRNAs directly impairs CFTR activity by binding to a specific target" in Table 2.

94) "combat the CFTR" does not seem to make sense in "This study suggests a new therapeutic approach of using mRNA-miRNA interaction to combat the CFTR" in Table 2. How can one "combat" a transmembrane regulator?

95) Please replace "isomiRNA" with "isomiRNAs" (line 515).

96) Please change "miR126-3p has as a target the interleukin 6 cytokine family signal transducer (IL6ST) which" to "The target of miR126-3p the interleukin 6 cytokine family signal transducer (IL6ST)" (line 521).

97) Please specify the key molecular player of "trans-signalling" mentioned in "Interestingly, it has been recently shown that a CF bronchial cell line is hyper-responsive to both the classic IL-6 and trans-signalling as compared to non-CF counterparts" (line 523).

98) Please change "The IL-6" to "IL-6-mediated" (line 525).

99) Please replace "an increase of" with "increased" (line 528).

100) Please change "would possibly exert" to "possibly exerts" (line 530).

101) Please replace "CFLD, similarly the" with "CFLD. Similarly," (line 531).

102) Please change "miR-223-3p has as a target IL6ST" to "The target of miR-223-3p IL6ST" (line 540).

103) Please replace "testifies" with something like "exemplifies" (line 544).

104) Please change "with" to something like "found within" (line 545).

105) Please replace "controls also" with "controls" (line 552).

106) Please change "in" to "in the" (line 556).

107) Please replace "MiR-155-5p" with "miR-155-5p" (line 561).

108) Please change "models (ALI cultures of bronchial, nasal, and nasal polyp cells)" to "models, ALI cultures of bronchial, nasal, and nasal polyp cells" (line 566).

109) Please replace "indicate" with "indicated" (line 568).

110) Please change "within" to "within the" (line 574).

111) Please replace "AMs" with "AM" (line 576).

112) Please change "MiR-126-3p" to "miR-126-3p" (line 586).

113) Please replace "miR-223-3p (verified by miRNet and TargetHumanScan databases)" with "miR-223-3p, which was verified by miRNet and TargetHumanScan databases" (line 587).

114) Please change "polyphosphate-5-Phosphatase" to "polyphosphate-5-phosphatase" (line 588).

115) Please replace "miR-155-5p (also validated using TargetHumanScan and miRNet
databases)" with "miR-155-5p, which was also validated using TargetHumanScan and miRNet
databases" (line 589).

116) Please change "MiR-145-5p" to "miR-145-5p" (line 591).

117) From "Grey edges: background/predicted interactions in the full network" (line 602) is unequivocally clear whether the authors mean to say "background and predicted interactions" or "background or predicted interactions"? Please correct.

118) Please justify "Personalised medicine is the next big frontier in human dis-" to the left (line 639).

119) Please provide reference for "Along this line, a gender gap exists in CF, whereby females are at a clinical disadvantage as compared to males" (line 641).

120) Please replace "signalling and" with "signalling," (line 646).

121) Please change "migration" to "migration," (line 646).

122) Please replace "healthy
controls" with "HC", "range:1–6" with "range: 1–6", "years;" with "years," in Table 3.

123) Please format "miRNA microarray profiling" using font size consistent with the rest of the text in Table 3.

124) Please format "Eleven miRNAs differentially expressed between CF and HC samples. The overexpressed miRNAs included hsa-miR-486-5p, 3 family members of let-7 (hsa-let-7b-5p, hsa-let-7c-5p, and hsa-let-7d-5p), hsamiR-103a-3p, and other miRNAs, while hsa-miR-598-3p was underexpressed in CF" using font size consistent with the rest of the text in Table 3.

125) Please format "Two significantly differentially regulated miRNAs in male versus female samples, miR-885-5p and miR-193a-5p" using font size consistent with the rest of the text in Table 3.

126) Please define abbreviation for "HC" in the footnote to Table 3.

127) Please change "Abbreviations: EBC, exhaled breath condensate; EV" to "EV" (line 664).

128) Please replace "FEV1" with "FEV1%" (line 664).

129) Please change "FVC" to "FVC%" (line 664).

130) Please replace "controls; PE, pulmonary exacerbation" with "controls" (line 665).

131) Please change "Distribution" to "distribution" (line 667).

132) From "For miR-155-5p, the chronic inflammatory characteristic of the CF airway microenvironment activates NF-κB signalling, driving its up-regulation in epithelial cells and neutrophils; increased miR-155 suppresses SHIP1, thereby enhancing PI3K/Akt activation and contributing to the excessive production of pro-inflammatory mediators such as IL-8" (line 671) is not entirely clear what the authors mean by "miR-155 suppresses SHIP1"? Are they referring to the suppression of SHIP1 expression?

133) Please replace "up-regulation" with "upregulation" (line 673).

134) Please change "neutrophils; increased" to "neutrophils. Increased" (line 673).

135) Please replace "miRNAs" with "miRNA" (line 679).

136) Please change "MiR-223-3p" to "miR-223-3p" (lines 684, 685).

137) Please replace "MiRNA" with "miRNA" (line 713).

138) Please change "Project" to "project" (line 746).

Round 5

Reviewer 1 Report

Comments and Suggestions for Authors

Carbone and Sajid et al. have scrutinized the vast immunological landscape of the roles that micro RNAs (miRNAs) play in cystic fibrosis (CF). In the following order, they discuss miRNAs of the airway epithelia, lung macrophages, and neutrophils in relation to CF, the inherent complexity of the emerging CF miRNomes, miRNA content in CF pulmonary fluids, as well as general miRNA compartmentalization with the emphasis put on downstream molecular targets and the underpinning immune signaling pathways. The scope of the review is sufficiently long and is accompanied by 4 tables and 2 intriguing figures containing original datasets. As such, the comprehensive review by Carbone and Sajid et al. sheds new and important insights into the biology of non-coding RNAs on the backdrop of the cystic fibrosis lung disease.

1) Please change "transcription" to "and transcription" (line 23).

2) Please replace "fuelling" with "thereby fuelling" (line 49).

3) Please change "Pseudomonas" to "and Pseudomonas" (line 52).

4) Please replace "remodelling" with "remodelling" (line 58).

5) Please change "chitin sensing" to "the sensing of chitin" (line 120).

6) Please replace "AEC" with "AECs" (line 148).

7) Please change "of" to "of the" (lines 278, 581).

8) Please replace "was" with "is" (line 278).

9) Please change "levels and" to "levels, as well as" (line 287).

10) Please replace "miR-126" with "miR-126 let to" (line 297).

11) Please change "by the" to "by" (line 301).

12) Please replace "obtain a" with "obtain" (line 308).

13) Please provide reference for "However, similar results obtained with immortalised CF AECs strongly suggested that miRNA expression was related to AECs" (line 309).

14) Please change "to" to "to the" (line 314).

15) Please replace "i.e. AECs" with "AECs" (line 318).

16) Please change "i.e. fibroblasts" to "fibroblasts" (line 319).

17) Please replace "[85], thus" with "[85]. Thus" (line 319).

18) Please change "with" to "with the" (lines 325, 627).

19) Please replace "of the" with "of" (line 325).

20) Please change "they do" to "they" (line 334).

21) Please replace "and reactive" with "as well as reactive" (line 347).

22) Please change "response, such" to "response such" (line 351).

23) Please replace "need" with "needs" (line 361).

24) Please change "signaling" to "signaling" (lines 373, 655, 666).

25) Please replace "behaviours in CF have" with "behaviour in CF has" (line 410).

26) Please change "to accumulation" to "to the accumulation" (line 456).

27) Please replace "TGF-β" with "TGF-β," (line 476).

28) Please change "[139], a" to "as a" (line 477).

29) Please replace "occurring during sterile and non-sterile lung inflammatory conditions and recruited from the blood" with "are recruited from the blood during sterile and non-sterile lung inflammatory conditions" (line 481).

30) From "Whether miR-636 overexpression can deregulate neutrophil functions such as, i.e. by decreasing the expression of IL1R1 and IKKβ proteins, as well as increasing the expression of RANK, as shown in CF AECs [79], has to be demonstrated" (line 505) is not clear whether the authors refer to mRNA or protein "expression of RANK"?

31) Please change "the expression of IL1R1 and IKKβ proteins" to "protein expression of IL1R1 and IKKβ" (line 506).

32) "the" could be replaced with "its" (line 515).

33) Please replace "AECs" with "AECs," (line 526).

34) Please change "NCF" to "non-CF" (4x), "its target gene TOM1 expression" to "the expression of its target gene TOM1", "41 dysregulated miRNA" with "41 dysregulated miRNAs", "CFTR- UTR3’ region" to "CFTR-UTR3’ region,", "3’ untranslated" to "UTR3’", "increase the the" to "increase" in Table 2.

35) Please define abbreviation for "UTR3’" in the footnote to Table 2.

36) Please change "miRNA; NCF, non-CF" to "miRNA" (line 532).

37) Please replace "IL-6 and" with "IL-6-mediated and the" (line 541).

38) Please change "to" to "to their" (line 542).

39) Please replace "miR-223-3p" with "miR-223-3p is" (line 558).

40) Please change "leading to reduced" to something like "thereby mitigating" (line 576).

41) Please replace "and particularly" with "and" (line 595).

42) Please change "using" to "using the" (line 607).

43) Please specify the "database" mentioned in "In the network tree, miR-181a-5p and miR101-3p do not display any interaction with any of the target genes of interest in this database network" (line 608).

44) Please replace "this database network", "this database", or "this" (line 609).

45) Please change "target" to "target the" (line 610).

46) Some of the blue circle nodes are not visible on the background mesh of blue edges in Figure 1. Please change their color to something like purple, red, or pink.

47) Please replace "highlighting target" with "target" (line 618).

48) From the legend to Figure 1 is not clear which database was used to construct the presented interaction network. Please fix.

49) It is not clear what the authors mean by "precise mRNAs" in "Blue edges: interaction among precise miRNAs and their specific targets (validated or top predicted targets)" (line 620)?

50) Please change "among" to "among the" (line 625).

51) Please replace "those" with "those miRNAs" (line 632).

52) The blue ellipses depicting altered miRNAs are not exactly vertically aligned in Figure 2. Please fix.

53) The yellow boxes depicting altered miRNA targets are not exactly vertically aligned in Figure 2. Please fix.

54) The green boxes depicting altered miRNA-regulated signaling pathways are not exactly vertically aligned in Figure 2. Please fix.

55) The blue ellipses depicting altered miRNAs are not exactly vertically aligned in Figure 2. Please fix.

56) The arrows next to the "miR-126-3p", "miR-233-3p", "miR-155-5p", "miR-181a-5p", "miR-101-3p" blue ellipses seem not to be precisely vertical in Figure 2. Please redraw.

57) The arrows next to the yellow "TOM-1", "IL6ST", "BACH1" boxes seem not to be precisely vertical in Figure 2. Please redraw.

58) The arrow next to the "NF-κB" green box
seems not to be precisely vertical in Figure 2. Please redraw.

59) Please change "hyper-inflammation" to "hyperinflammation" in Figure 2.

60) Please define the meaning of the blue ellipses, and yellow and green boxes in the legend to Figure 2.

61) Please replace "arrow" with "arrows" (line 641 2x).

62) Please change "Although" to something like "Although it is" (line 656).

63) Please replace "microRNA" with "miRNA" (line 662).

64) Please change "as a start to understand" to something like "a step towards understanding" (line 671).

65) Please replace "were" with "are" (line 673).

66) Please change "pediatric" to "paediatric" (lines 674, 677).

67) Please specify the type of "exacerbation" mentioned in "Since four peripheral and sputum miRNAs (let-7c, miR-16, miR-25-3p, and miR-146a) were the best predictive model of exacerbation, EV-miRNAs contained in peripheral blood or local samples (sputum but not EBC) might have a diagnostic potential of predicting exacerbation in pediatric CF" (line 674)?

68) Please replace "were" with something like "represented" (line 675).

69) Please change "EV-miRNAs contained" to "EV-miRNAs" (line 676).

70) Please replace "pediatric" with "paediatric", "3 family" with "family", "FEV1% predicted and FVC% predicted" with "the predicted FEV1% and FVC%" in Table 3.

71) Please change "controls" to "controls." (line 684).

72) Please replace "characteristic" with something like "milieu" (line 691).

73) Please change "signalling, particularly" to "signalling particularly" (line 699).

74) Please replace "works" with something like "studies" (line 717).

75) Please change "gauging" to something like "evaluating" (line 733).

76) Please replace "cells (AECs, monocyte, macrophages, and neutrophils)" with "cells" (line 754).

77) Please change "positioned" to "poised" (line 756).

Author Response

Dear Reviewer,

We would like to thank you for your thoughtful revisions and endorsement about the relevance of our article.

Since all academic revisions have been done, in agreement with the Editorial office the journal's editorial team will take care of the formatting problems and we will fix any issues in English language at the galley proof stage.

Our Best Wishes for a Happy New Year.

Massimo Conese